# Effects of psoriasis and psoralen exposure on the somatic mutation landscape of the skin

Sigurgeir Olafsson[1], Elke Rodriguez [2], Andrew R. J. Lawson [1], Federico Abascal [1], Axel Rosendahl Huber[3], Melike Suembuel[2], Philip H. Jones [1], Sascha Gerdes[2], Iñigo Martincorena [1], Stephan Weidinger[2], Peter J. Campbell [1] & Carl A. Anderson [1]✉

Somatic mutations are hypothesized to play a role in many non-neoplastic diseases. We performed whole-exome sequencing of 1,182 microbiopsies dissected from lesional and nonlesional epidermis from 111 patients with psoriasis to search for evidence that somatic mutations in keratinocytes may influence the disease process. Lesional skin remained highly polyclonal, showing no evidence of large-scale spread of clones carrying potentially pathogenic mutations. The mutation rate of keratinocytes was similarly only modestly affected by the disease. We found evidence of positive selection in previously reported driver genes *NOTCH1*, *NOTCH2*, *TP53*, *FAT1* and *PPM1D* and also identified mutations in four genes (*GXYLT1*, *CHEK2*, *ZFP36L2* and *EEF1A1*) that we hypothesize are selected for in squamous epithelium irrespective of disease status. Finally, we describe a mutational signature of psoralens—a class of chemicals previously found in some sunscreens and which are used as part of PUVA (psoralens and ultraviolet-A) photochemotherapy treatment for psoriasis.

Recent studies have advanced our understanding of the somatic mutation landscapes of phenotypically normal tissues, including blood[1], esophagus[2,3], colon[4–7], liver[4,8,9], urothelium[10,11], skin[12–15] and more[16–21]. These have shown that driver mutations, which provide cells with a competitive advantage over their neighbors, are common in non-neoplastic tissues and sometimes represent sizable fractions of all cells sampled.

Widespread replacement of wild-type cells with mutant clones can have functional consequences for the tissue, potentially contributing to common complex disease risk or influencing disease progression or response to treatment[22–24]. Exposure to environmental insults such as drugs or inflammation may influence somatic mutation landscapes and be correlated with disease. For example, azathioprine, which is used to treat several immune-related conditions, leaves a characteristic mutational signature on the genomes of exposed cells in both the skin[25] and colon[26].

We and others have recently shown how somatic mutation landscapes of affected tissues are altered profoundly in nonalcoholic fatty liver disease[8,9] and inflammatory bowel disease (IBD)[26–28]. In both conditions, disease results in increased mutagenesis and large-scale

replacement of wild-type cells with clones carrying somatic mutations in metabolic and immune-related pathways, respectively. In IBD, chronic inflammation drives the expansion of clones carrying somatic mutations affecting the toll-like receptor and interleukin-17 signaling pathways within the colonic mucosa[26–28].

Here, we further characterize the somatic mutation landscape of diseased tissues by studying chronically inflamed psoriatic skin. We use whole-exome sequencing (WES) of hundreds of microbiopsies of epidermis isolated from lesional and nonlesional skin of psoriasis patients to explore the extent to which somatic mutations affect the disease process in psoriasis. We also describe the mutagenic processes that are active in psoriatic skin, including the mutagenic effects of psoralen exposure, which is a part of commonly used phototreatment for psoriasis.

## Lesional skin of psoriasis patients is highly polyclonal

We recruited 111 psoriasis vulgaris patients between the ages of 18 and 88 years at the Department of Dermatology, University Hospital

[1]Wellcome Sanger Institute, Hinxton, UK. [2]Department of Dermatology and Allergy, University Hospital Schleswig-Holstein, Kiel, Germany. [3]Institute for Research in Biomedicine, Barcelona, Spain. ✉e-mail: carl.anderson@sanger.ac.uk

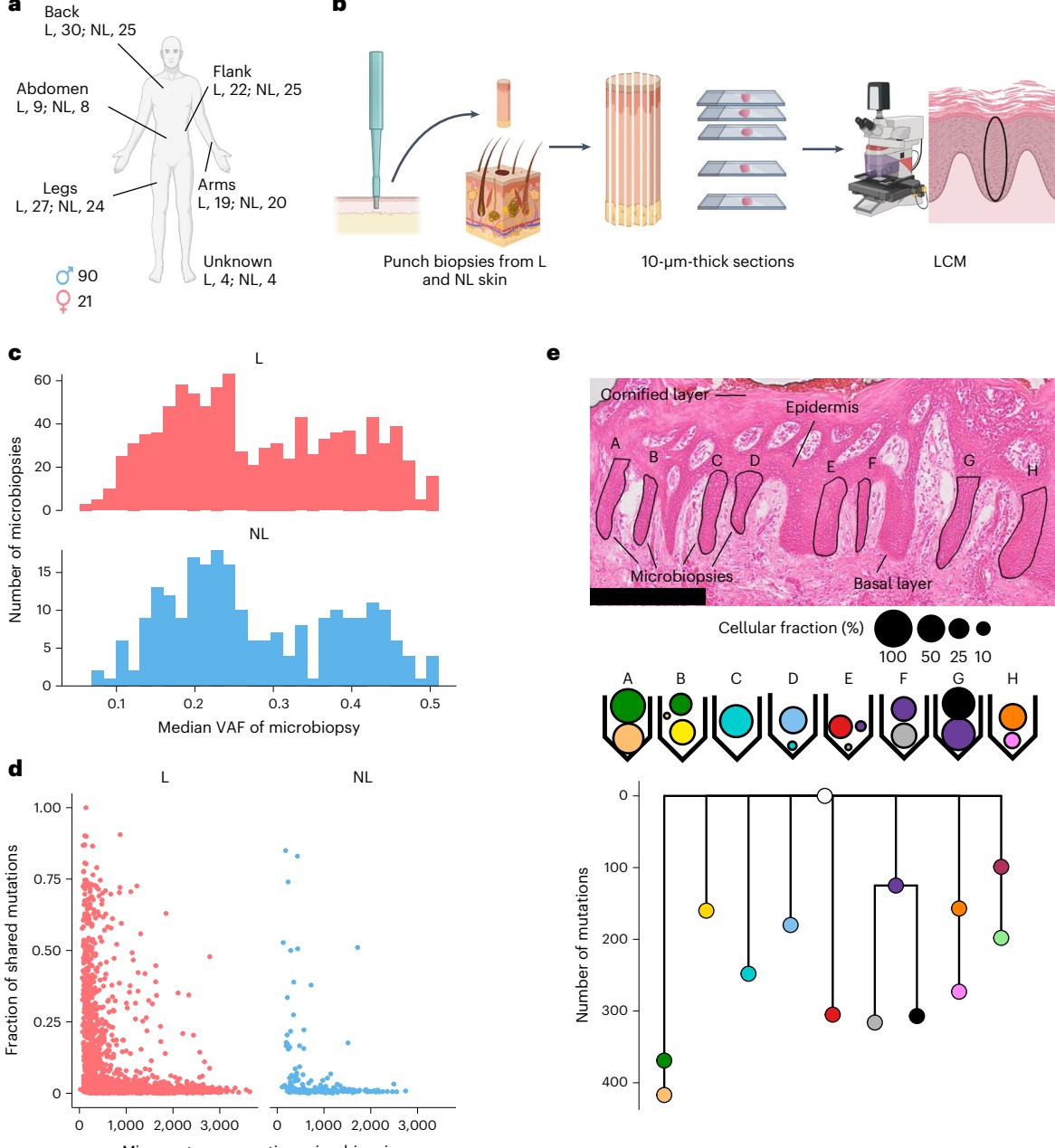

**Fig. 1 | Overview of the sampling strategy and clonal structure of the tissue.**
**a**, Anatomical locations of paired biopsies from lesional (L) and nonlesional (NL) skin. **b**, An overview of the sampling pipeline. Punch biopsies from L and NL skin taken under local anesthesia were sectioned and stained histologically. LCM was used to isolate several microbiopsies of epidermis from each skin biopsy. **c**, Histogram showing median VAFs of mutations called in microbiopsies from L and NL skin. The medians of the distributions are 0.26 for L skin and 0.25 for NL skin. **d**, For all microbiopsy pairs from the same skin biopsy, the figure shows the

fraction of mutations the members of the pair share in common as a function of the distance between the pair. No differences in mutation sharing are observed between L and NL skin biopsies. **e**, Top, histology image of a lesional biopsy (hematoxylin and eosin staining). Examples of microbiopsies of epidermis are highlighted. Scale bar, 250 µm. Middle, estimated cellular fractions of mutation clusters with VAF >1% found within the microbiopsies above. Bottom, phylogenetic tree for the patient based on the mutation clusters above. The rightmost branch is from a NL biopsy not shown on the histological image.

Schleswig-Holstein, Kiel, Germany. Patients donated paired punch biopsies from lesional ($n = 111$) and adjacent nonlesional ($n = 106$) skin (Fig. 1, Extended Data Fig. 1a–c, Supplementary Note 1 and Supplementary Table 1; Methods). Laser capture microdissection (LCM) was used to isolate 1,182 microbiopsies of epidermis from this material (946 lesional (80%); 236 nonlesional (20%)) (Fig. 1 and Supplementary Tables 2 and 3; Methods). Healthy skin is maintained by a single layer of proliferative stem cells that line the basal membrane; their progeny stratify vertically to the outer layers of epidermis. Most microbiopsies

correspond to <0.01 mm² of skin surface (Extended Data Fig. 1) and are the progeny of a small number of basal stem cells. Assuming a diameter of 9 µm and vertical stratification of cells from basal to cornified layer, we estimate that a median of 30 basal stem cells gave rise to the cells of the microbiopsies.

DNA extracted from each microbiopsy was whole-exome sequenced to a median on-target depth of 56× and somatic substitutions, indels and structural variants were called (Supplementary Table 4 and Supplementary Notes 1 and 2; Methods). We called a median of

430 single-base substitutions, 39 double-base substitutions and three indels per microbiopsy. We sequenced 18 near-replicate samples, where the same histological features visible on adjacent histological slides were dissected twice and sequenced independently. A median of 89% of substitutions and indels were called in both samples (Extended Data Fig. 1g; Methods), providing a quantitative estimate of the sensitivity of the mutation calling pipeline (albeit excluding mutations with very low variant allele fractions (VAFs) in both samples).

We found that even our small microbiopsies of epidermis rarely comprised fully clonal populations of cells, with most microbiopsies containing a mixture of clones. The VAFs of microbiopsies dissected from lesional and nonlesional skin were near identical (Median $VAF_{lesional} = 0.26$, Median $VAF_{nonlesional} = 0.25$; Fig. 1d) and clones extending over many microbiopsies or large areas of the skin punch biopsies were rarely observed. The fraction of microbiopsy pairs more than 500 μm apart that shared more than 10% of their mutations was identical (4.1%) between lesional and nonlesional skin (Fig. 1e). This indicates that the hyperproliferation of keratinocytes associated with flare-ups of psoriasis has minimal effect on the vertical spread of mutant clones.

## Mutagenic processes in psoriatic skin

The oligoclonal composition of the microbiopsies means their mutation counts are not representative of the per cell mutation counts in the skin. There is also a risk of counting the same mutation event several times when a clone extends across more than one microbiopsy. We performed Bayesian clustering of mutations based on their VAFs in different microbiopsies and used the statistical pigeonhole principle to derive a phylogenetic relationship between the clusters[8,9] (Fig. 1e; Methods). Each tip of the phylogenetic trees represents a clone and we performed all subsequent analyses at the level of clones rather than the level of microbiopsies.

To determine which mutational processes have been active in the skin over the lifetime of our psoriasis patients, we used a Bayesian hierarchical Dirichlet process (Methods) to extract mutational signatures for each mutation clone and compared these with the COSMIC reference signatures[29,30] (Fig. 2a, Extended Data Figs. 2 and 3 and Supplementary Table 3). Unsurprisingly, the most abundant signature is single-base substitution (SBS) SBS7b, which has been attributed to ultraviolet (UV) exposure. This signature accounts for 80% of the mutations in the dataset. We also found the UV-related signature SBS7c, but it accounts for only 0.14% of the mutations. In agreement with previous studies[15,20], we observed large variation in UV-associated mutation burden between cells 1–2 mm apart in the tissue. Of the patients in our cohort, 26 have a history of phototreatment with UV-B (Extended Data Fig. 1b,c), but we did not find those individuals to have a significantly higher burden of SBS7b and SBS7c than patients without history of UV-B treatment, possibly due to lack of power (Supplementary Note 3).

The second most prevalent signature, accounting for 10.7% of the mutations in the dataset, is not listed in COSMIC (v.3.2). It is characterized by T > A, T > C and T > G mutations at TpA sites (Supplementary Table 5). This is consistent with the known mutagenic effects of psoralens[31–33]—a class of chemicals that form a part of PUVA (psoralens and UV-A) phototreatment, which is used to treat psoriasis and other diseases of the skin. We found no evidence that other common treatments, including methotrexate or topical steroid use, cause somatic mutations in the skin (Supplementary Note 3).

The remaining prevalent signatures identified are the result of cell-intrinsic mutational processes. One is a mix of the clock-like SBS1 and SBS5 signatures that are found ubiquitously in normal tissues, including proliferative and postmitotic tissues[21,34,35]. These signatures tend to be correlated in a tissue and could not be separated by our model. We also found a small number of samples exhibiting the mutational signatures of apolipoprotein B mRNA-editing catalytic polypeptide-like (APOBEC) activation (SBS2 and SBS13), which have been found occasionally to be present in a range of normal tissues[10,34,35].

While the APOBEC signatures account for only 0.5% of the mutations in the dataset as a whole, 22 clones had over 50 exonic mutations attributed to SBS2/SBS13, accounting for up to 28% of mutations in the exposed clones. This indicates that members of the APOBEC family of proteins can be activated occasionally in the skin, as shown across a range of other normal tissues[10,26,34,35].

To assess whether psoriasis is associated with difference in the mutation burden of the skin, we fitted linear mixed effects models (LMMs) to estimate the independent effects of age, disease duration and anatomical location of the biopsies while accounting for the fact that samples are not independent but display a hierarchical structure by patients and biopsies (Supplementary Note 4; Methods). As mutations attributed to psoralen exposure do not accumulate linearly with age, we subtracted the estimated burden of the psoralen signature from the total mutation burden of each clone before fitting the models. We estimate that the mutation burden of keratinocyte clones, excluding psoralen mutations, increases by 14.6 mutations per exome per year (14.6 (10.1–19.0, 95% confidence interval (CI)), $P = 4.3 \times 10^{-9}$, LMMs; Supplementary Note 4). This is a much higher burden than that of other normal tissues, which range around 0.2–0.4 mutations per exome per year[1–10,34,35].

Lacking information about the frequency or severity of psoriasis flare-ups, we next tested whether disease duration, which we have used previously as a proxy for inflammation exposure in IBD[26], is associated with mutation burden of keratinocyte clones (Supplementary Note 2; Methods). We did not find a significant effect of disease duration on the total mutation burden (–0.60 mutations per year (–3.4 to 2.2, 95% CI, $P = 0.67$, likelihood ratio test of LMMs; Fig. 2d)). However, the large variation in the burden of the UV-related SBS7b reduces statistical power to detect a disease effect on the total mutation burden. In our previous work on IBD-affected colonic mucosa, we found that the disease was associated with accelerated mutagenesis by the cell-intrinsic signatures SBS1 and SBS5 (ref. 26). We therefore fitted a model using only the mutation burden attributed to SBS1/5 and found that psoriasis increases the mutation burden of these signatures by 0.16 mutations per exome per year of disease duration (0.038–0.29, 95% CI, $P = 0.012$, likelihood ratio test of LMMs; Supplementary Note 4). We estimate the age effect of SBS1/5 to be 0.65 mutations per year (0.49–0.81, 95% CI, $P = 2.5 \times 10^{-12}$, LMM; Supplementary Note 4). Some of the mutation clusters consisted of groups of mutations with VAFs too low for the pigeonhole principle to be incontrovertible. The calculations above assume that, in such cases, the mutations all derive from a single subclone. In Supplementary Notes 2 and 4, we describe the effects of repeating the calculations using only branches that represent single clusters.

## Mutational signature of psoralen exposure

Psoralens are a class of linear furanocoumarins, which are polycyclic aromatic compounds found naturally in many crops, including citrus fruits and figs. Their synthetic forms are administered as part of phototherapy (PUVA treatment, psoralens + UV-A) of psoriasis as well as other severe skin diseases such as eczema, vitiligo and graft-versus-host disease. Psoralens were also included as tanning activators in some tanning lotions and sunscreens until a limit on their use was imposed by European safety regulations in 1996. Psoralens are used as mutagens in molecular biology research and both PUVA treatment and the use of psoralen-containing sunscreens are known to increase the risk of skin cancers[36–39]. Upon irradiation, the psoralen molecule binds to thymine forming a monoadduct. Further exposure to light can then cause the molecule to form an interstrand crosslink between thymines on opposite strands at TpA sites.

In our whole-exome data, we found a signature characterized by T > A, T > C and T > G mutations at TpA sites (Fig. 3a), consistent with the known mutagenic effects of psoralens[31–33]. Out of the 111 donors, 24 showed evidence of the signature. As with UV-light-related mutations, proximal cell clones sometimes show large variation in the number

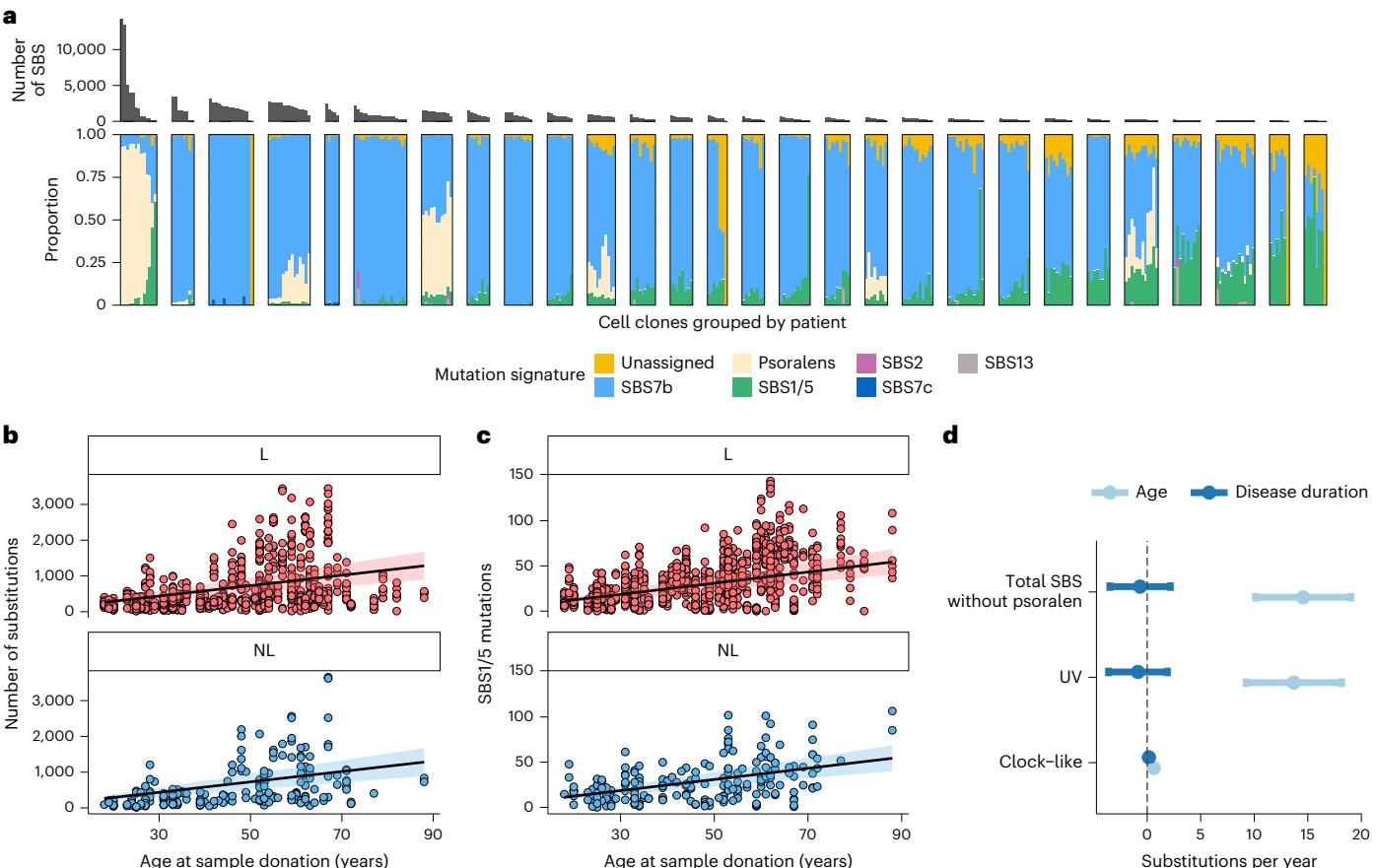

**Fig. 2 | Mutational signatures and mutation burden in lesional and nonlesional skin. a**, Number of SBS in each clone and the relative contribution of each mutational signature. Each bar represents a tip of the phylogenetic tree for that patient (a clone). Clones are grouped by patient, and patients are ordered by mutation burden of the clone with the most mutations. For clarity, only samples taken from the leg are shown. See Extended Data Fig. 3 for other anatomical sites. Patient 34 in the leftmost panel is an outlier, showing extremely high mutation burden of the psoralen signature. **b**, SBS burden not attributed to psoralens as a function of donor age at the time of sampling. **c**, Mutation burden of the clock-like SBS1/5 as a function of age. **d**, Comparison of fixed effects for age and disease duration in linear mixed effect models predicting the total mutation burden after subtracting psoralen-related mutations, the UV-mutation burden and the burden of the clock-like signatures SBS1 and SBS5. The *y* axes in **b**–**d** refer to the exonic mutation burden (note different scales of the *y* axes); shaded regions show 95% CI of the age-effect estimate.

of psoralen-related mutations, with some showing extremely high mutation burdens. In the most extreme case, over 12,000 exonic mutations attributed to psoralen were found in two related cell clones from patient 34, who has a history of extensive PUVA treatment (>200 sessions). These clones did not show a higher-than-expected burden of double-base substitutions or indels. Across the dataset in general, the indel mutation spectrum is identical (cosine similarity >0.99) between clones with high and low burden of the substitution psoralen signature (Extended Data Fig. 4). The mutation spectra of double-base substitutions in both groups are also dominated by the UV-related CC > TT mutations (DBS1), suggesting that the mutagenic effects of psoralens are limited to single-base substitutions.

We found that including the number of PUVA cycles as a covariate in a model of the psoralen signature mutation burden significantly improved the fit of the model ($P = 1.2 \times 10^{-6}$, likelihood ratio test of LMMs) and, although only the group with >200 cycles of PUVA had a significantly higher psoralen mutation burden than those without history of PUVA treatment (616 extra (exonic) mutations per clone, 407–826, 95% CI), we observed a trend for increased mutation burden with the number of PUVA cycles (Fig. 3b). However, the presence of the psoralen signature did not perfectly correlate with history of PUVA treatment. Out of the 24 patients where the signature was observed, 11 have no documented history of PUVA treatment (Fig. 3b),

suggesting that the signature may also occur through other kinds of psoralen exposure. The median age of these 11 individuals is 55 years and 10 were born before 1980. We hypothesize that some may have used psoralen-containing lotions/sunscreens in the past, the effects of which persist to this day. The signature was not observed in eight patients with PUVA treatment histories (Supplementary Table 1). Of these, seven had either low or an unknown number of treatment cycles. It is possible that these caused an insufficient number of mutations for us to detect using our WES approach or that treatment may in some cases have been applied locally to a different body site than was sampled for this study.

We did not find evidence of the psoralen signature in any of 468 cases of cutaneous melanoma samples in the cancer genome atlas (TCGA)[40]. However, one patient with basal cell carcinoma (BCC) in the Hartwig Medical Foundation (HMF) cohort[41], HMF003357A, did show clear evidence of the signature (Fig. 3a). A HMF study coordinator reviewed this patient's medical records and found that they have a history of psoriasis and PUVA treatment. Although anecdotal, this example provides an orthogonal validation of the signature in an independent cohort and suggests that, whereas psoralen exposure seems to be a rare source of mutagenesis in sporadic skin cancers, it can come to dominate the mutational spectrum in patients who have received PUVA treatment, even greatly surpassing the effects of UV from sun exposure.

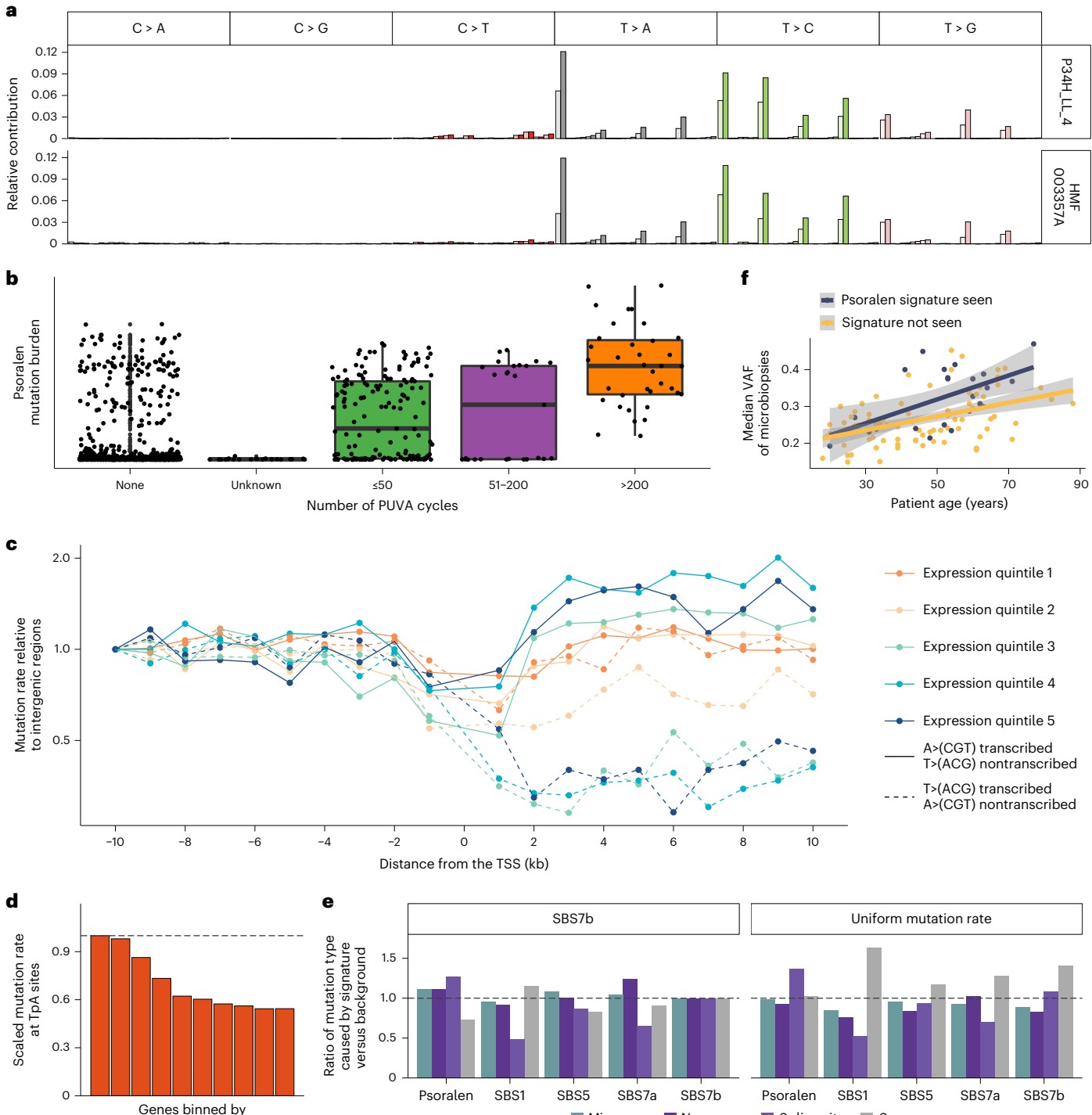

**Fig. 3 | Characterization of the mutational signature of psoralen exposure.**
**a**, Upper, 96-class mutation profile of one of the outlier samples from patient 34, who has a history of PUVA treatment and shows extremely high burden of the psoralen signature. The lighter shades represent mutated pyrimidine bases on the transcribed strand and the darker shades the untranscribed strand. Lower, mutation profile of a BCC sample from the Hartwig medical foundation cohort. This patient has a history of psoriasis and PUVA treatment. **b**, Psoralen mutation burden as a function of the number of PUVA cycles the patient has received. Horizontal lines represent the medians and edges of the boxes the first and third quartiles. Whiskers extend to 1.5× the interquartile range. **c**, Mutational densities in the vicinity of the TSS of genes by expression level quintiles in sun-exposed

skin in the GTEx dataset. Mutation rate on the transcribed strand drops in the transcribed region of genes, reflecting TCR. In contrast, the mutation rate on the untranscribed strand is increased in the transcribed region, compared with intergenic regions, reflecting TCD of this strand. The effects become more pronounced with increasing expression. **d**, Mutation rate in ten bins of ascending gene expression. Higher expressed genes have lower mutation rates. **e**, Expected number of mutations in different annotation classes in 15 genes linked with SCC normalized by the number of mutations under a uniform mutation rate or SBS7b. **f**, Median VAFs of all microbiopsies from a patient as a function of patient age. Shaded regions show 95% CIs.

To further characterize the mutagenic effects of psoralen exposure in vivo, we performed whole-genome sequencing (WGS) of 16 microbiopsies from three patients who showed clear evidence of the signature in the whole-exome data and two of whom have histories of PUVA treatment (Fig. 3 and Supplementary Table 2). We found an effect of sequence context that extended beyond the trinucleotide context, with sites that contain the double palindrome ApTpApT being most frequently mutated (Extended Data Fig. 4b).

We found a strong effect of transcription on psoralen-related mutagenesis in the WGS data, with 1.72 times more mutations occurring on the untranscribed strand than on the transcribed on average (Fig. 3a). This is due partially to the effects of transcription-coupled repair (TCR), which removes mutations from the transcribed strand, but the asymmetry is further compounded by transcription-coupled damage (TCD) to the complementary nontranscribed strand (Fig. 3c and Supplementary Note 5). Although both TCD and TCR occur, the impact of TCR is about twice that of TCD, with the result that the mutation rate at TpA sites drops with increasing gene expression (Fig. 3d).

In addition to the effects of transcription, there was an effect of replication on psoralen-related mutagenesis. The leading strand accumulates 9% more mutations compared with the lagging strand, and late-replicating regions consistently showed higher mutation rates than early replicating regions (Extended Data Fig. 4c,d and Supplementary Note 5).

High burden of the psoralen signature is also associated with clonal expansions of keratinocytes. The median VAFs of all microbiopsies dissected from a patient increases with the age of the patient, but individuals showing evidence of the psoralen signature have higher median VAFs after accounting for age ($P = 0.0017$, likelihood ratio test; Fig. 3f). Psoralens are cytotoxic and may enable clonal expansions through the elimination of competitor clones. Exposure may also result in a higher number of driver mutations and thus higher fitness.

Among genes reported to be under positive selection in squamous cell carcinomas (SCC) or normal skin (listed in Methods), we counted all possible mutations that would be predicted to result in synonymous, missense, nonsense or splice-site mutations. We then compared the fraction of mutations in each annotation class that would be expected for the psoralen signature and four other signatures commonly found in the skin (Fig. 3e) with the number of mutations that would be expected under a uniform background mutational process and under SBS7b—the dominant mutational process in the current study. This analysis suggested that psoralens are less likely to result in synonymous mutations in known driver genes compared with other mutational processes affecting the skin, and may be especially likely to cause splice-site mutations. We also note that a number of canonical hotspot mutations in known skin cancer genes are caused by T > A, T > C or T > G mutations at TpA sites and list these in Supplementary Table 6.

## Positive selection in lesional and nonlesional skin

Previous studies have shown that a large fraction of cells in the normal skin carry mutations in genes that are recurrently mutated in keratinocyte cancers[12–15]. The recurrent cycles of inflammation, hyperproliferation and remission that characterize psoriasis could create an environment where different selection forces drive clonal expansions, for example, by propagating mutations that render cells resistant to the cytotoxic effects of inflammation[23,24,27,42].

We used dNdScv software[43] to assess the ratio of nonsynonymous to synonymous mutations (dN/dS) after accounting for sequence context and regional differences in mutation rate across the genome (Supplementary Table 7 and Supplementary Note 6). We first carried out an exome-wide screen for recurrently mutated genes in all microbiopsies and found nine genes that passed correction for multiple testing (Fig. 4a and Extended Data Fig. 5). Mutations in five of these (*NOTCH1*, *FAT1*, *PPM1D*, *TP53* and *NOTCH2*) have been shown previously to be under positive selection in normal skin in studies that have used deep

sequencing of targeted gene panels[12,15]. Four additional genes (*GXYLT1*, *CHEK2*, *ZFP36L2* and *EEF1A1*) that were not a part of the targeted panels or have not been previously reported reached significance in the current study. To explore whether mutations in these four genes are uniquely under positive selection in lesional skin, we repeated the selection analysis using only samples from lesional skin or nonlesional skin. No additional genes reached significance in these restricted analyses but *ZFP36L2* and *GXYLT1* did not reach significance when nonlesional microbiopsies were excluded (Supplementary Note 6). *ZFP36L2* and *GXYLT1* showed nominal significance ($P < 0.05$) in nonlesional skin, suggesting that mutations in these genes are not specifically selected for under conditions of inflammation and that our failure to detect significant evidence of positive selection is probably driven by lower power due to the reduced sample size.

We next carried out a pathway-level dN/dS analysis, searching for enrichment of missense and truncating variants across 11 genesets that were defined a priori because of their relevance in either keratinocyte cancers or psoriasis pathology (Fig. 4c; Methods). We observed a strong enrichment of mutations in genes previously reported to be recurrently mutated in normal skin or in SCC. Genes reported to be recurrently mutated in BCC showed a weaker enrichment. No enrichment was seen for either missense or nonsense mutations in any of the other pathways implicated in psoriasis pathogenesis. Together with the gene-level analysis, this suggests that somatic mutations in keratinocytes are unlikely to play a role in the pathogenesis of psoriasis.

Based on the VAFs of individual mutations and the volume of the microbiopsies in which they were detected, we estimated for each patient the fraction of cells that carry a mutation in each gene (Fig. 4d; Methods). We tested whether the fraction of mutant cells was different between lesional and nonlesional skin using a Wilcoxon test for paired measurements but found no differences that were significant after correction for multiple testing. This further supports the hypothesis that the selection forces operating in psoriasis are the same as those operating in the skin of people without psoriasis.

## Discussion

We have used WES of microbiopsies of epidermis derived from 111 patients with psoriasis vulgaris to study the effects of this chronic disease on the somatic mutation landscape of the skin. Our analyses suggest that psoriasis is unlikely to have a major impact on the lateral spread of keratinocyte clones in the skin, that disease effects on the mutation burden are modest and that similar mutations confer selective advantage on cells in psoriasis as in healthy skin. Psoriasis is associated with modestly increased risk of keratinocyte cancers[44,45] but our results support the view that this increase in risk may be due predominantly to the effects of treatment rather than features of the disease per se.

Although the effects of UV light (SBS7) unsurprisingly dominate in the skin, other mutational processes also affect keratinocytes and may be affected by disease. There is evidence from other diseases that inflammation is associated with increased mutation burden of the clock-like signatures SBS1 and SBS5, which represent cell-intrinsic processes and are found in all cells of the body[34,35]. Here we found that disease duration of psoriasis, our best proxy for inflammation exposure, is associated with increased mutation burden of SBS1/5. We estimate that psoriasis disease duration is associated with an additional 0.16 mutations per exome per year, which is approximately a quarter of the age effect of these signatures. However, this estimate is dependent on certain assumptions about the clonal structures of the microbiopsies (Supplementary Note 2) and more data will be needed to corroborate this potential effect. If the burden of SBS1/5 is increased in psoriasis, it will result in only a modest increase in the total mutation burden of keratinocytes, as SBS1/5 explains a small fraction of the total mutation burden of keratinocytes compared with UV-related signatures.

We discovered that mutations in four genes that have not been reported before in studies of normal epidermis are under positive

 

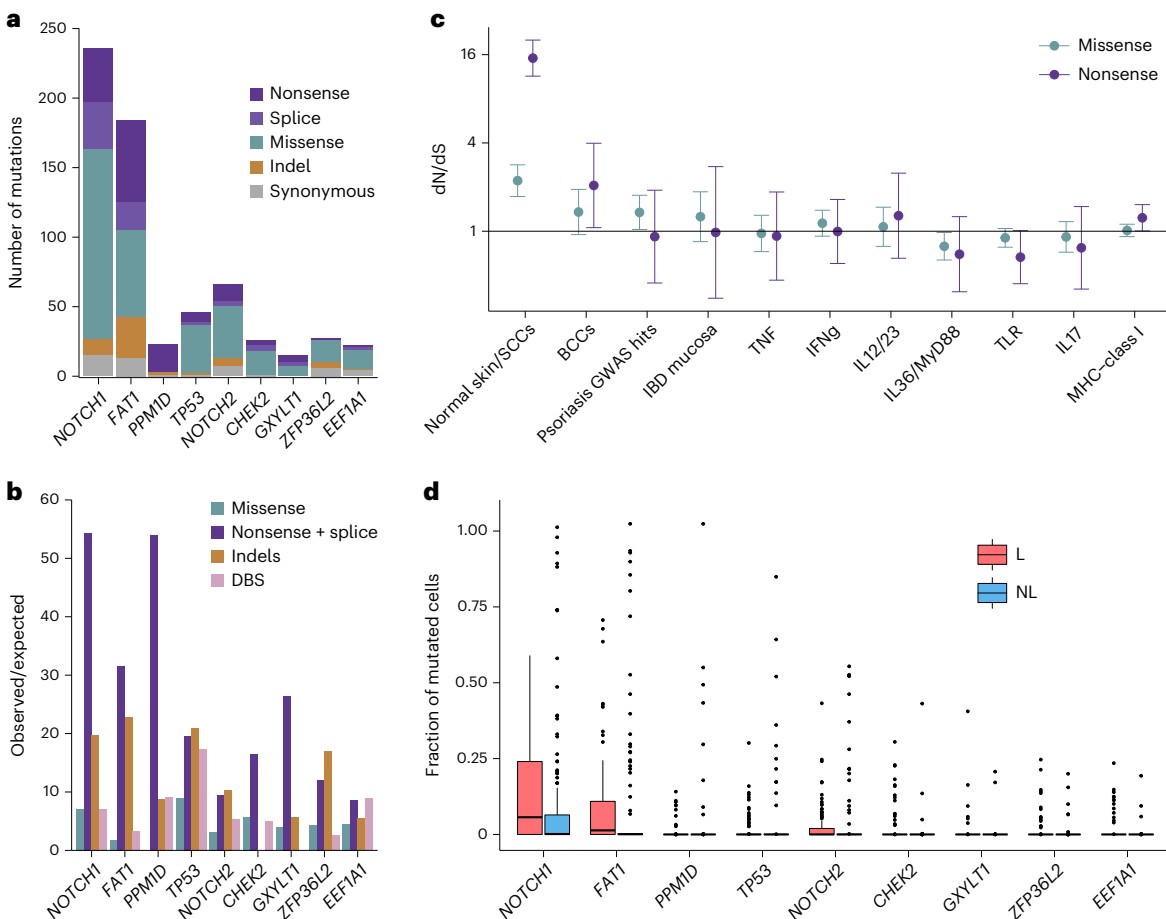

**Fig. 4 | Positive selection in psoriatic skin. a**, Number of mutations in genes found to carry a significant excess of mutations by functional annotation. **b**, Observed-to-expected ratios for different mutation classes in each gene. **c**, Pathway-level dN/dS[43] (pentanucleotide model) for genes known to be recurrently mutated in keratinocyte cancers and genesets relevant to psoriasis pathogenesis. TNF, tumor necrosis factor; IFN, interferon; IL, interleukin; TLR,

Toll-like receptor; MHC, major histocompatibility complex. Error bars show 95% CIs of the dN/dS estimates for each pathway in the center. **d**, Comparison of the fraction of mutated cells between lesional and nonlesional skin. Lines show the medians of each group while the lower and upper hinges show the 25th and 75th percentiles; whiskers extend from the hinges to 1.5× the interquartile range.

selection in our dataset. We hypothesize that nonsynonymous mutations in these genes are under positive selection in squamous epithelia in general and are not specific to psoriasis or selected as a consequence of the inflammatory environment (Supplementary Note 2). Mutations across all nine genes that reached significance in the study seemed to be present at a similar frequency across lesional and nonlesional skin from psoriasis patients, further suggesting they are not a specific characteristic of lesional skin.

We have described a mutational signature that we believe to be the result of psoralen exposure. While psoralens have been known to be mutagenic for decades, we add considerable detail to our understanding of this mutagen in humans in vivo. Although the correlation between the signature and clinical history of PUVA treatment was not perfect, the etiology of the signature is supported by existing experimental evidence showing that psoralens cause mutations at TpA sites[31–33]. While psoralen exposure in the context of psoriasis is particularly likely to occur during PUVA-therapy, other possible means of exposure include the past use of sunscreen or tanning lotions containing psoralens or even the consumption of furocoumarin-rich foods. It has been hypothesized that orally ingested furocoumarins increase risk of skin cancers[46–49]. We looked for evidence of the psoralen signature both in TCGA melanoma samples and in skin cancers from the HMF and found evidence of the signature only in a psoriasis patient

with a history of PUVA treatment. This does not rule out a mutagenic effect of furocoumarins from diet, but suggests that they are rarely a principal source of mutations in the populations from which these cohorts are drawn. We propose that testing for a relationship between furocoumarin consumption and the psoralen mutational signature in sun-exposed skin offers a mechanistic way to test the hypothesis that furocoumarin-rich foods cause skin cancers.

Genome-wide association studies have revealed both disease specific and widespread sharing of disease mechanisms between immune-mediated diseases. Many pathways and disease mechanisms are shared across several autoimmune conditions but the extent to which these relationships manifest in the somatic mutation landscape of affected tissues is not clear. Comparison with our earlier work on IBD shows that, although they are both Th17-mediated chronic inflammatory diseases of epithelial tissues, and share several genome-wide association study loci in common[50–52], IBD and psoriasis have very different effects on the somatic mutation landscapes of affected tissues. In IBD, mutations in immune-related genes are under positive selection and these may play a role in the pathogenesis of the disease and/or enable cells to escape the cytotoxic effects of IL-17 (ref. 27). Mutations in genes of the IL-17 pathway are not positively selected in psoriasis, underscoring the seemingly different role of this cytokine in these two diseases. It is worth noting that while IL-17 inhibitors work well for psoriasis,

clinical trials of these drugs in IBD have either shown the drugs to not be effective or that they worsen the disease[53,54]. Importantly, our data does not exclude a role for somatic mutations in immune cells in psoriasis pathogenesis[55]. Emerging data from other autoimmune diseases suggests that somatic mutations may sometimes enable lymphocytes to bypass tolerance checkpoints, resulting in expansion of autoreactive clones[56,57]. Determining whether similar mechanisms are at play in psoriasis is an important direction of future work.

## Online content

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

## Methods

This study complies with all relevant ethical regulations. All donors gave informed consent for genetic research of the material and the study was approved by the research ethics committee of Christian-Albrechts University in Kiel (A100/12), the National Health Service (NHS) Research Ethics Committee (Yorkshire and The Humber—South Yorkshire Research Ethics Committee, REC ID 20/YH/0244, IRAS ID 286843) and by the Wellcome Trust Sanger Institute Human Materials and Data Management Committee (approval number 20/0085).

### Statistics and reproducibility

This is a descriptive study with no interventions. No randomization of patients was applied. All dependent variables were generated computationally and no blinding was applied. Sample size was not predetermined. We excluded from the whole study six microbiopsies suspected of being sample swaps and/or contaminated with external DNA. A total of 101 microbiopsies were excluded only from the analysis of structural variants either due to the inability of the algorithm to find an optimal solution or if the goodness-of-fit for the optimal solution was <90%.

### Human tissue attainment and processing

**Tissue donation and fixation.** Round punch biopsies, 4 mm in diameter, from lesional and adjacent nonlesional skin were donated by psoriasis patients presenting to the Department of Dermatology, University Hospital Schleswig-Holstein, Kiel between 2017 and 2019. Biopsies were fixed in RNAlater (ThermoFisher, catalog no. AM7021) upon collection following the manufacturer's instructions and frozen. Patients were not compensated for participation in the study.

**LCM of epidermis.** Skin biopsies were processed using a Tissue Tek VIP 6 AI tissue processor (Sakura Finetek) and embedded in paraffin. The blocks were sectioned into 10-μm-thick sections using an Accu-Cut SRM 200 microtome (Sakura Finetek). Sections were fixed to 4 μm PEN membrane slides (Leica, catalog no. 11600288). Slides were stained with hematoxylin and eosin and imaged using a NanoZoomer 2.0-HT slide scanner (Hamamatsu Photonics). Samples of epidermis were dissected from this material using LCM microscopy (Leica, catalog no. LMD7000). Cells were lysed using Arcturus PicoPure DNA extraction kit (Applied Biosystems, catalog no. KIT0103) according to the manufacturer's instructions.

### DNA sequencing

**Whole-exome sequencing.** A total of 1,182 microbiopsies (Supplementary Table 2) from 111 individuals were whole-exome sequenced on Illumina NovaSeq 6000 machines using 150 bp paired-end reads and the Agilent SureSelect Human All Exon V.5 bait set (catalog no. S04380110). Paired-end reads were aligned to the human reference genome (build hg38) using BWA-MEM[58]. PCR duplicates were marked using biobambam[59] and duplicate statistics were calculated using Picard (v.1.131) (http://broadinstitute.github.io/picard/). Sample contamination estimates were calculated using VerifyBamID (v.1.1.3)[60] and the on-target coverage was calculated using Samtools (v.1.11) depth command, considering only reads with base quality and mapping quality >30. The median on-target coverage for all samples was 56× (range, 18–190) for the dataset as a whole but 58× and 46× for microbiopsies from lesional and nonlesional skin, respectively. The difference in coverage is due to the greater volume of the microbiopsies from lesional skin (Extended Data Fig. 1), which results in greater library complexity and translates to lower PCR duplicate rate and higher coverage. We cut columns of cells and the thicker epidermis of lesional skin results in a greater sample volume for the same surface area covered.

**Whole-genome sequencing.** WGS was performed on 16 microbiopsies from three donors (patients 18, 21 and 34) who showed the signature of PUVA exposure in the WES data. The sequencing of these samples is further described in Supplementary Note 1.

### Somatic mutation calling

**Mutation calling in whole exomes.** Substitutions were called using CaVEMan (v.1.15.1) (Cancer Variants through Expectation Maximization) (https://cancerit.github.io/CaVEMan/)[61]. Mutations were called against an unmatched normal with the copy number options set manually to 10 and 2 for the mutant and wild-type copy numbers, respectively. The samples were compared against a normal panel consisting of 75 unrelated normal samples to remove common SNPs. Mutations were further filtered if the reads reporting the mutations had a median alignment score <140 or if >50% of the reads were clipped.

All mutations passing these filters in any sample from a donor were next genotyped in all samples from that donor. We used the bam2R() function of the deepSNV[62] R package (v.1.40.0) to generate pileups of all sites mutated in any sample using only mapped reads that had base quality and mapping quality >30 and which were mapped in a proper pair, were not PCR duplicates, were the primary alignment and which passed platform quality check (SAM Flags 3847, see https://broadinstitute.github.io/picard/explain-flags.html). After these filters, we required a coverage of at least four times at the site and at least three reads reporting the alternate allele in at least one sample from a donor to call a mutation.

Adjacent substitutions called in the same sample were merged into a double-base substitution call if the number of reads reporting the reference and alternative alleles was not significantly different (Fisher test).

Indels were called using cgpPindel (v.3.5.0) (https://github.com/cancerit/cgpPindel)[63] using the same unmatched normal sample that was used to call substitutions. We generated pileups of the indel calls in the same way as described above for substitutions and required a coverage of at least four times and at least three reads reporting the alternate allele in at least one sample from a donor to call a mutation as before.

**Binomial filtering of somatic mutation calls.** To filter rare germline variants not removed by the comparison with the normal panel we applied an exact binomial test of the number of reads reporting each mutation[26]. Heterozygous germline variants are expected to be present at a VAF of 0.5 in every sample from a patient. For each mutation, we compared the number of reads reporting the reference and alternate alleles across all samples from that patient. We tested the hypothesis that the read counts for the variants were drawn from a binomial distribution with a probability of success of 0.5, or 0.95 for mutations on the sex chromosomes in men. We applied Benjamini–Hochberg correction for multiple testing and excluded mutations with $q > 10^{-3}$. We also used binomial filtering to remove erroneous mutation calls. Recurrent sequencing artefacts will be distributed randomly across samples and can be modeled as being drawn from a binomial distribution. In contrast, true somatic mutations will have a high VAF in some samples whilst being completely absent from others. The latter are best represented by a beta-binomial with a high overdispersion. For every mutation call, we calculated the maximum likelihood overdispersion parameter ($\rho$) in a grid-based way (ranging the value of $\rho$ from $10^{-6}$ to $10^{-0.05}$)[26]. Calls with $\rho < 0.1$ were filtered as probably artifactual.

The sensitivity of the mutation calling postfiltering was estimated at 89% as described in the Supplementary Note 1.

**Structural variant calling in the exome data with allele-specific copy number analysis of tumors.** Structural variants were called using allele-specific copy number analysis of tumors[64] (https://github.com/VanLoo-lab/ascat) as further described in Supplementary Notes 1 and 2.

### Identification of SNV clusters by a hierarchical Dirichlet process

We implemented a nonparametric Bayesian hierarchical Dirichlet process (HDP) to cluster autosomal SBSs with similar VAFs. The full mathematical and implementation details of the model are described

in a previous publication[8]. Briefly, clones of cells are present across different microbiopsies and this manifests as clusters of mutations that are found at similar VAFs within a microbiopsy. For every mutation, we have two vectors, one containing the number of reads reporting the alternate allele in each microbiopsy and another containing the total sequencing depth at each microbiopsy. We assume that each mutation can be assigned to exactly one cluster but the number of clusters is unknown. We aim to estimate the number of clusters present across all the microbiopsies dissected from a patient, the location of each cluster in the n-dimensional VAF hypercube and the allocation of mutations to each cluster.

We model the data using an N-dimensional Dirichlet process clustering model, where the distribution of clone sizes and numbers follows a Dirichlet process. This has the advantage that there is no need to prespecify the number of clusters present. Instead, mutations are moved around the clusters and, in each sampling iteration, there is a defined probability that a mutation will initiate a new cluster that was not present in previous iterations. Clusters can also cease to exist if all member mutations are assigned to other clusters. Thus, the number of clusters varies throughout the sampling chain. Details of the implementation are provided in the Supplementary Note 1 and in the code accompanying the manuscript.

### Inference of phylogenetic trees

Each cluster of single-base-substitutions identified by the N-dimensional Dirichlet process algorithm represents a branch of the phylogenetic tree for that patient. We applied the statistical piegeonhole principle to infer phylogenetic relationships between clusters. Given clusters A and B, if the combined mutant cell fraction (CF) of both is >100% (VAF > 0.5) in the same microdissections and B consistently shows a lower CF than A, then that is strong evidence that B is nested within A, that is, mutation cluster B represents a subclone of clone A. If the combined mutant CF is ≤100%, only weak evidence of nesting exists. If B is found at a higher VAF than A in some microdissections but at lower VAF in others, the clusters are interpreted as being independent clones without nesting. We treated each tip of the phylogenetic tree for each patient as a clone. The length of the branches from the root (germline) was used in the mutation burden calculations.

Some clusters are present at VAFs too low for the pigeonhole principle to be incontrovertible (that is, combined cell fraction <100%). In these cases, there is a risk that the cluster does not represent a single clone as assumed above, but a mixture of clones present at similar VAFs. We reconstructed the phylogenetic trees after pruning away branches where there is doubt about the validity of the pigeonhole principle. We retained nested clusters only if the sum of the cellular fraction estimates exceeds 1. Un-nested clusters (that is, branches of the phylogenetic tree consisting of a single cluster) were retained if the median VAF of the cluster is >0.3.

### Mutational signatures

**Extraction of mutational signatures.** To extract mutational signatures and estimate the exposure of each signature, we used a second hierarchical Dirichlet process[65], as implemented in the HDP (v.0.1.5) R package (https://github.com/nicolaroberts/hdp). The data were organized into a tree structure where the root contained all the mutations in the dataset. This node had, as children, one node that represented the most recent common ancestor of all the patients and frozen pseudo-count nodes for signatures that are to be used as priors in the model. The pseudonodes contained 10,000 pseudocounts each. We used COSMIC signatures 1, 5, 2, 13, 7a, 7b, 7c, 7d, 17a, 17b, 18 and 38 as priors. During the Dirichlet process, mutations from the dataset may join the pseudocount clusters, but the pseudocounts are frozen such that they are unable to leave the initial cluster. The patient ancestor node had, as children one node for each of the patients, and each patient node had as children one node for each branch of the phylogenetic tree. The

parameters used are provided in Supplementary Note 1 and the code accompanying the manuscript.

The model extracted nine signature components in addition to the unassigned component (Extended Data Figs. 2 and 3). Among these were components corresponding to signatures SBS7b, SBS1/5, SBS2, SBS7c and SBS13, all of which were included as priors in the model. One component did not correspond to any COSMIC signature but is characterized by mutations at TpA sites and accounted for just under 11% of the mutations in the dataset. This is the signature we attribute to psoralen exposure, which, in the context of psoriasis, is likely to occur during treatment with psoralens and high-dose UV-A (PUVA) (see the main text). Finally, three additional components were extracted: unknown components N1–3 in Extended Data Fig. 2. These together accounted for 2.6% of the mutations in the dataset. They may represent individual variation in repair of UV-damage or they may be artefacts of the signature extraction model. We do not have sufficient confidence in these components to draw conclusions from them and have added them to the unassigned component for subsequent analyses.

**Characterization of the psoralen signature.** From the whole-exome data, we identified a number of samples that showed a large number of mutations at TpA sites, consistent with the known mutagenic effects of psoralens[31–33]. To enable further characterization of this signature, we selected 16 microbiopsies from patients showing clear evidence of psoralen exposure for WGS.

To visualize the trinucleotide and pentanucleotide spectrums associated with psoralen exposure, we used the R package MutationalPatterns[66] (v.3.4.0) together with BSgenome (v.1.60.0). To calculate the transcriptional strand bias, we used the gene definitions from the R package TxDb.Hsapiens.UCSC.hg38.knownGene (v.3.13.0) and the strand_occurrences() and strand_bias_test() functions from MutationalPatterns.

To test for TCD, we carried out a similar analysis to that used originally to describe TCD in liver cancers[67]. We divided protein-coding genes into quintiles by ascending expression in sun-exposed skin from the GTEx dataset[68] (v.8). We extracted the transcriptional start site (TSS) and the strand of each gene from Gencode (v.27) and defined ten 1 kb bins upstream and downstream of the TSSs. We pooled T > (ACG) and A > (CGT) mutations at TpA or ApT sites from all whole-genome sequenced samples. If the gene is on the (−) strand, the transcribed strand is the reference and we counted the number of T > (ACG) mutations overlapping each 1 kb bin. If the gene is on the (+) strand, the transcribed strand is the complement of the reference and we counted the number of A > (CGT) mutations. This was reversed for the untranscribed strand. We observed a drop in the mutation rate on the transcribed strand upstream of the TSSs, indicating TCR. However, we also found an increased mutation burden on the untranscribed strand, indicative of TCD of this strand. To test the statistical significance of the increased mutation burden, we fit two linear models with and without a parameter indicating whether each position was upstream or downstream of the TSSs and used a likelihood ratio test to test whether the fit of the model was improved. Figure 3c shows the mutation rate in each 1 kb bin relative to the −10 kb bin—the intergenic bin furthest from the TSS.

To test the effect of gene expression levels on PUVA mutagenesis we again used expression data from sun-exposed skin from the GTEx dataset[68] (v.8). We split protein-coding genes into ten equally sized bins by ascending levels of expression. We used the Bedtools[69] intersect function (v.2.18) to count the number of mutations overlapping genes in each bin. Figure 3d shows the relative mutation rate in each bin compared with the lowest expression bin. We also looked for an effect of the replication timing and replication strand as described in Supplementary Notes 1 and 5.

To assess the potential functional effects of the psoralen signature relative to other main signatures in the skin (Fig. 3f), we used the

context_potential_damage_analysis() and signature_potential_damage_analysis() functions of the MutationalPatterns package in R. Considering only the genes previously reported to be under selection in squamous cell carcinomas and/or normal skin, we counted the number of trinucleotide changes expected to give rise to the different types of mutations (synonymous, missense, nonsense or splice site). For each of the main signatures in the skin (Psoralen, SBS1, SBS5, SBS7a and SBS7b), we normalized the expected fraction of mutations falling in each annotation class by the fractions that would be expected given either a uniform mutation rate or SBS7b. We note that, whereas this analysis gives a hint of how damaging a signature might be, it is based only on a trinucleotide mutational context and other important features, such as gene expression, strandedness or the extended nucleotide context, are not taken into account. The genes we used in this analysis are *AJUBA*, *ARID2*, *ASXL1*, *CASP8*, *CDKN2A*, *FAT1*, *KMT2D*, *NOTCH1*, *NOTCH2*, *NOTCH3*, *PPM1D*, *RB1*, *RBM10*, *TP53* and *TP63*.

To see whether psoralen exposure was associated with increased levels of clonal spread after controlling for age, we estimated the median VAF for microbiopsies dissected from each patient and regressed this against the age of the patient in a linear model. We then fitted a second linear model which also included as covariate whether the psoralen signature was seen in that patient. We used a likelihood ratio test to see whether this improved the fit of the model (Fig. 3f).

### Mutation burden estimation

**Linear mixed effect models.** We used linear mixed effect models to compare the mutation burdens of lesional and nonlesional skin and test for an effect of disease duration. We used as response variables the estimates of the mutation burdens of SBS7 and SBS1/5 after correcting for VAF and coverage as described in Supplementary Notes 1 and 3. The models include fixed effects for age and the anatomical location of the biopsy from which the clone is derived. Cell clones from the same biopsy are likely to have correlated levels of UV-exposure and some correlation is also likely to exist between biopsies taken from the same patient. To model this, we include random effects for patient and biopsy, with the effect of biopsy being nested within that of the patient. We provide a mathematical description of the models used in Supplementary Note 1 and implement the models in R in Supplementary Note 4.

### Selection and driver analyses

**Exome-wide driver discovery.** We used the dNdScv (v.0.0.1.0) software[43] (https://github.com/im3sanger/dndscv) to identify genes enriched in nonsynonymous mutations, indicative of positive selection. We first calculated dN/dS ratios across all coding genes using mutations identified in either lesional or nonlesional skin, identifying positive selection of mutations in nine genes (Fig. 4). We next estimated global dN/dS values for groups of genes belonging to 11 genesets defined a priori as described in Supplementary Note 1.

**Fraction of mutated cells.** We compared the fraction of cells that carry mutations in any of the nine genes that showed a significant enrichment of mutations between lesional and nonlesional skin. We calculated the fraction of mutated cells separately in lesional and nonlesional biopsies from each individual by multiplying twice the VAF of each mutation in each microbiopsy by the volume of the microbiopsy and dividing that by the total volume of microbiopsies dissected from that patient. For clones that carry more than one mutation in the same gene, we counted only the mutation with higher VAF.

### Reporting summary

Further information on research design is available in the Nature Portfolio Reporting Summary linked to this article.

## Data availability

Raw sequencing data are available in the European Genome-phenome Archive (EGA) using study ID EGAS00001004882 and dataset ID EGAD00001011265. Intermediary and supporting files, including mutation calls, mutational cluster assignments, phylogenetic trees, histological images, spatial relationship matrices and more are available in a Mendeley data repository (https://doi.org/10.17632/phvh82vd9g.1). The mutation calls from the TCGA project can be obtained by using the TCGAmutations package in R (https://github.com/PoisonAlien/TCGAmutations). Access to the mutation calls of the HMF cohort can be obtained using the request forms found at https://www.hartwig-medicalfoundation.nl/en/. Source data are provided with this paper.

## Code availability

Code supporting the main analyses of the manuscript is provided as R-markdown supplementary analysis files. Custom scripts documenting the mutation filtering and clustering pipelines, signature extraction and more are publicly available at https://github.com/Solafsson/somaticPsoriasis.

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

## Acknowledgements

We thank the staff of the Wellcome Sanger Institute's Sample Management, Sequencing and Informatics teams for their contribution to the study. Special thanks go to Y. Hooks for histological processing and sectioning of the skin biopsies and to D. Plowman and S. Leggett for practical assistance in sample and data management. We also thank the German psoriasis patients who donated tissue samples for the study. This research was funded by the Wellcome Trust (Grant numbers 206194 and 108413/A/15/D, Sanger Institute

core funding, managed by C.A.A.] and by the Innovative Medicines Initiative 2 Joint Undertaking under grant agreement number 821511 (BIOMAP), managed by S.W.

## Author contributions

S.O., P.J.C. and C.A.A. designed the study with contributions from E.R., S.W., P.H.J. and I.M. E.R., S.G. and S.W. consented patients for the study, collected clinical metadata and took and fixed the tissue biopsies. S.O. stained and imaged the tissue sections and performed LCM of microbiopsies of epidermis. S.O. performed the calling and quality control of substitutions and indels. A.R.J.L. performed the calling of structural variants. A.R.H. identified the individual showing the psoralen signature in the HMF dataset. SO performed all statistical and bioinformatics analyses with contributions from A.R.J.L., F.A., I.M., P.J.C. and C.A.A. S.O. wrote the initial draft of the manuscript and all authors contributed to the interpretation of the findings and the final version of the manuscript.

## Competing interests

S.W. has received institutional research grants from Sanofi Deutschland GmbH, LEO Pharma and Pfizer, and performed consultancies and/or lectures for AbbVie, Almirall, Boehringer, Eli Lilly, Galderma, Kymab, Leo Pharma, Regeneron, Sanofi-Genzyme and Novartis. C.A.A. has received consultancy or lecture fees from Genomics plc, BridgeBio and GlaxoSmithKline. Since completing this work, S.O. has become an employee of deCODE Genetics, a subsidiary of Amgen. S.G. has been an advisor and/or received speakers' honoraria and/or received grants and/or participated in clinical trials of the following companies: AbbVie, Affibody AB, Akari Therapeutics Plc, Almirall-Hermal, Amgen, Argenx BV, Aristea Therapeutics, Biogen Idec, Bioskin, Bristol-Myers Squibb, Boehringer-Ingelheim, Celgene, Dermira, Eli Lilly, Galderma, Hexal AG, Incyte Inc., Janssen-Cilag, Johnson & Johnson, Klinge Pharma, Kymab, Leo Pharma, Medac, MSD, Neubourg Skin Care GmbH, Novartis, Pfizer, Pierre Fabre, Principia Biopharma, Regeneron Pharmaceutical, Sandoz Biopharmaceuticals, Sanofi-Aventis, Trevi Therapeutics and UCB Pharma. M.S. has received speakers' honoraria and/or participated in clinical trials of the following companies: AbbVie, Affibody AB, Almirall-Hermal, Amgen, Anaptys Bio, Argenx BV, Biogen Idec, Boehringer-Ingelheim, Celgene, Dermira, Eli Lilly, Hexal AG, Incyte Inc., Janssen-Cilag, Kymab, Leo Pharma, Novartis, Pfizer, Regeneron Pharmaceutical, Sanofi-Aventis, Trevi Therapeutics and UCB Pharma. I.M. and P.J.C. are academic cofounders and consultants of Flagship Labs 86. The remaining authors declare no competing interests.

## Additional information

**Extended data** is available for this paper at https://doi.org/10.1038/s41588-023-01545-1.

**Correspondence and requests for materials** should be addressed to Carl A. Anderson.

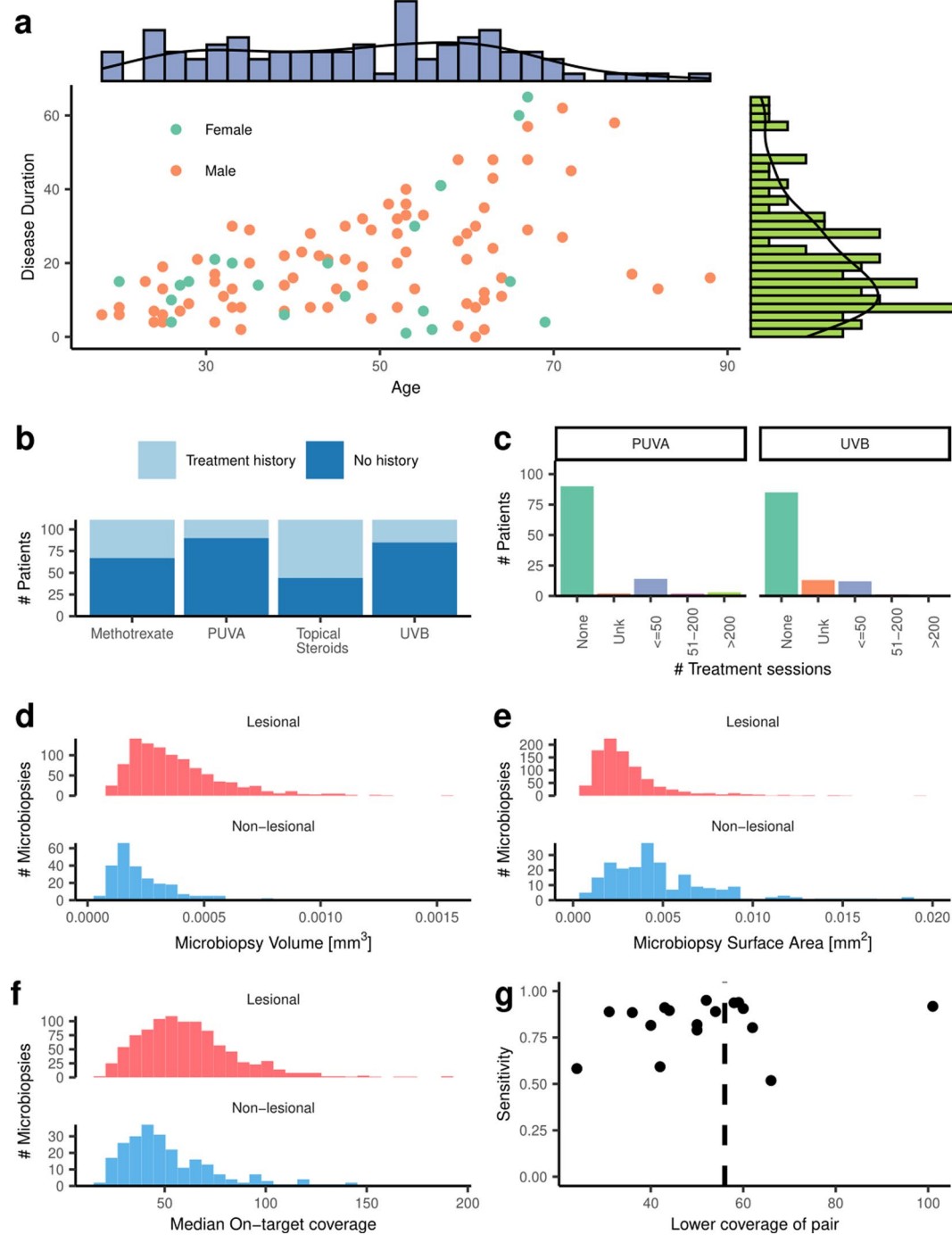

**Extended Data Fig. 1 | Patient and microbiopsy characteristics. a)** Time since psoriasis diagnosis as a function of age broken down by sex of the donor. **b)** Fraction of patients who have a history of any of four common treatments for psoriasis. **c)** The number of treatment cycles for two common phototreatments: Psoralens + UV-A (PUVA) and UV-B phototreatment. Unk = Number of treatments unknown. **d)** Histograms showing the volume of the microbiopsies by type.

**e)** Histograms showing the estimated surface area of the skin each microbiopsy covers. **f)** Median on-target coverage for each microbiopsy. The medians for each type are 58X and 46X for the lesional and non-lesional microbiopsies, respectively. **g)** Sensitivity estimates derived by sequencing 18 near-replicate samples of epidermis where the same histological features were visible on adjacent sections.

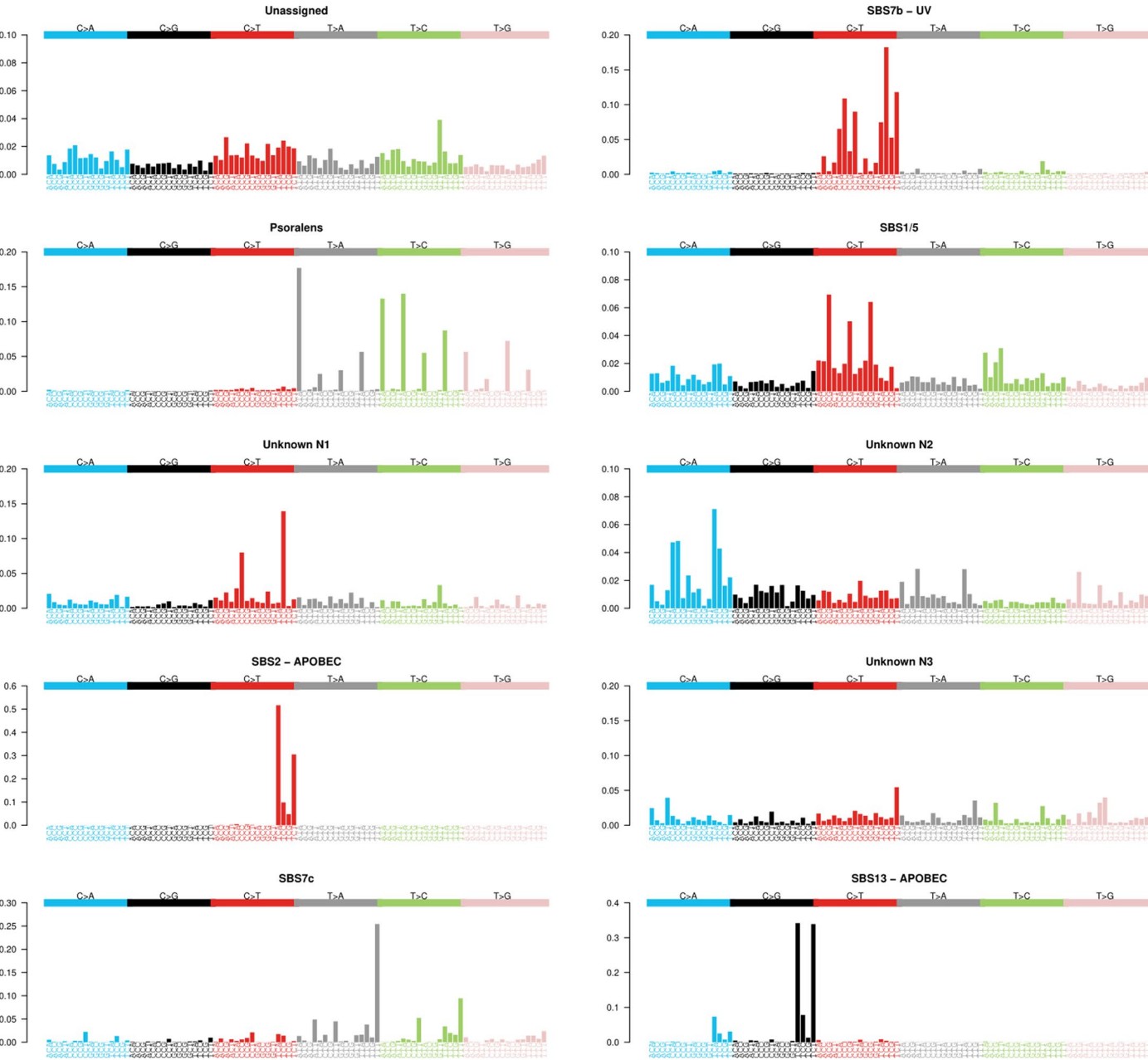

**Extended Data Fig. 2 | Mutational signature components extracted using a hierarchical Dirichlet process.** Apart from the Unassigned, components are ordered by the fraction of mutations assigned to each component in the dataset.

Three unknown components, N1-N3, were extracted that together account for 2.6% of the mutations. These have been merged with the Unassigned component in Main Fig. 2.

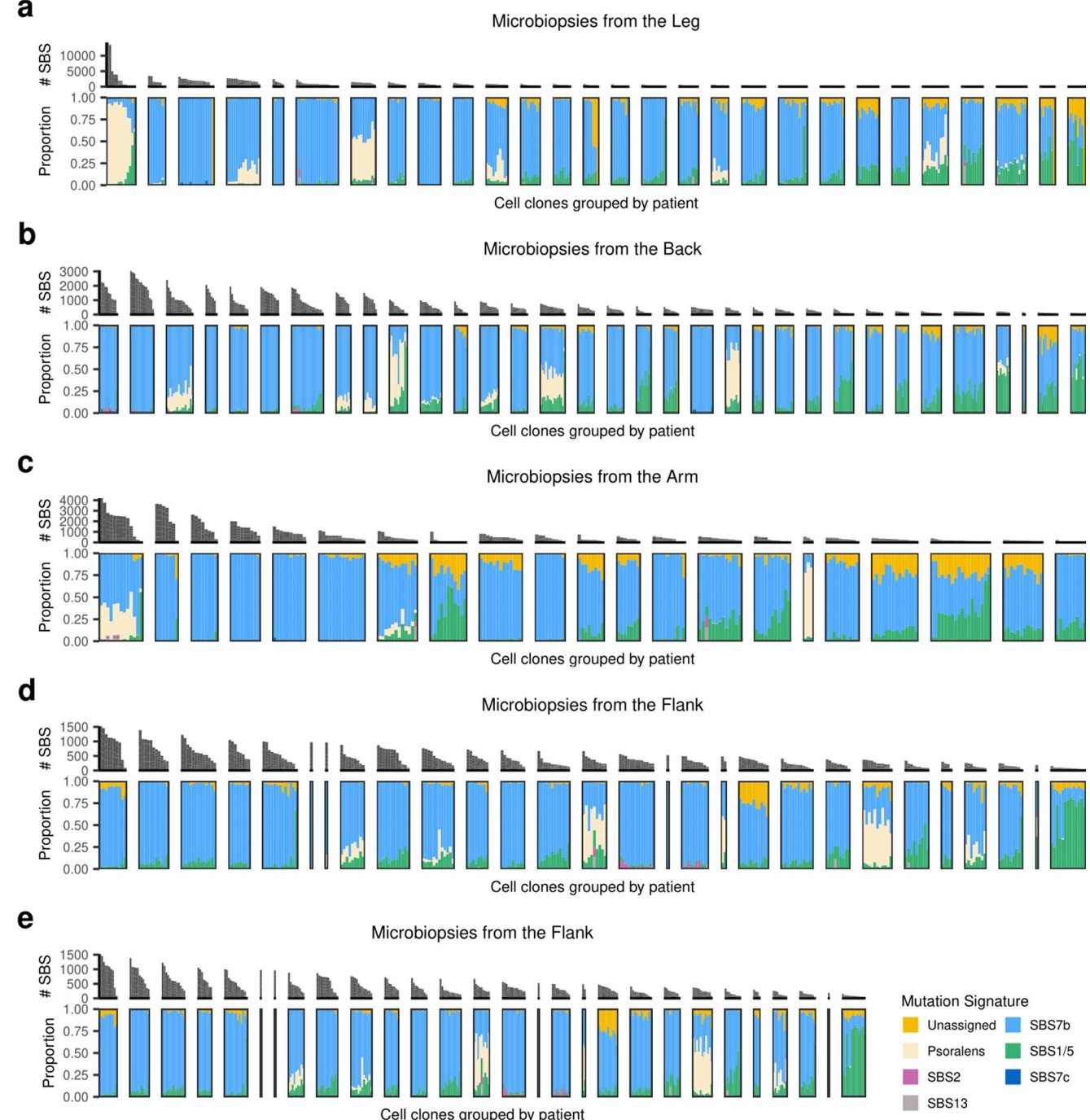

**Extended Data Fig. 3 | Mutational signatures across all cell clones by anatomical location of the skin microbiopsies.** The outlier, patient 34, is the leftmost patient that donated a sample from the leg.

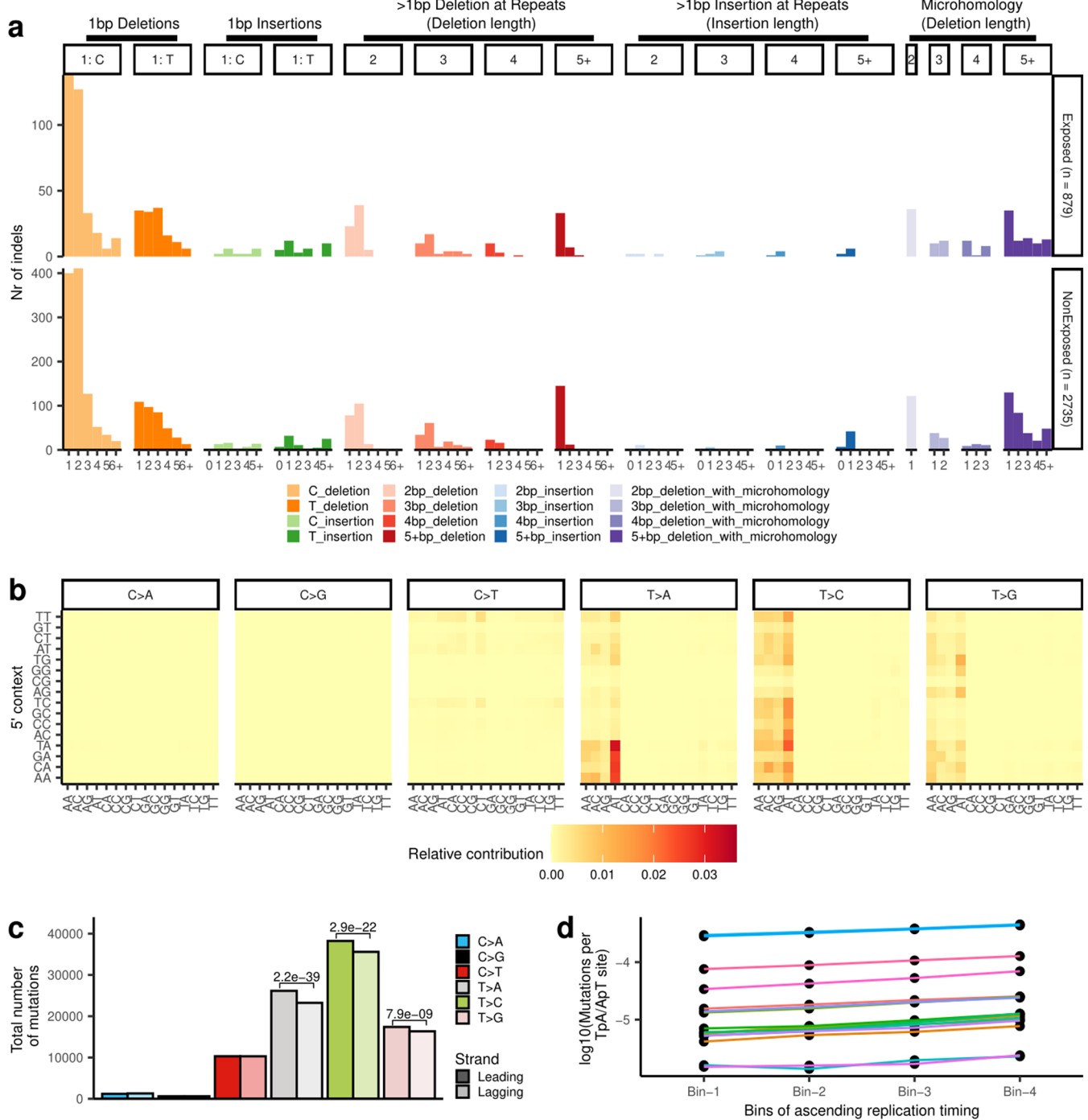

**Extended Data Fig. 4 | Further characterization of the psoralen signature.**
**a**) A comparison of the indel mutation spectra between samples with high and low burden of the psoralen substitution signature. The spectra are identical (cosine similarity >0.99), indicating that psoralens don't cause indels. *n* is the number of mutation events. **b**) A heatmap showing the effect of sequence context on psoralen-mutagenesis extends beyond the trinucleotide model, with bases 2 bp either side of the mutated base affecting the mutation probability.

**c**) The effect of replication direction of psoralen mutagenesis in whole-genome sequenced samples. More mutations are observed on the leading versus lagging strand. P-values are from a two-sided Poisson test and are not corrected for multiple testing. **d**) The effect of replication timing on psoralen mutagenesis. Across all samples that underwent whole-genome sequencing the mutation rate is greater in late-replicating regions as compared with early replicating regions.

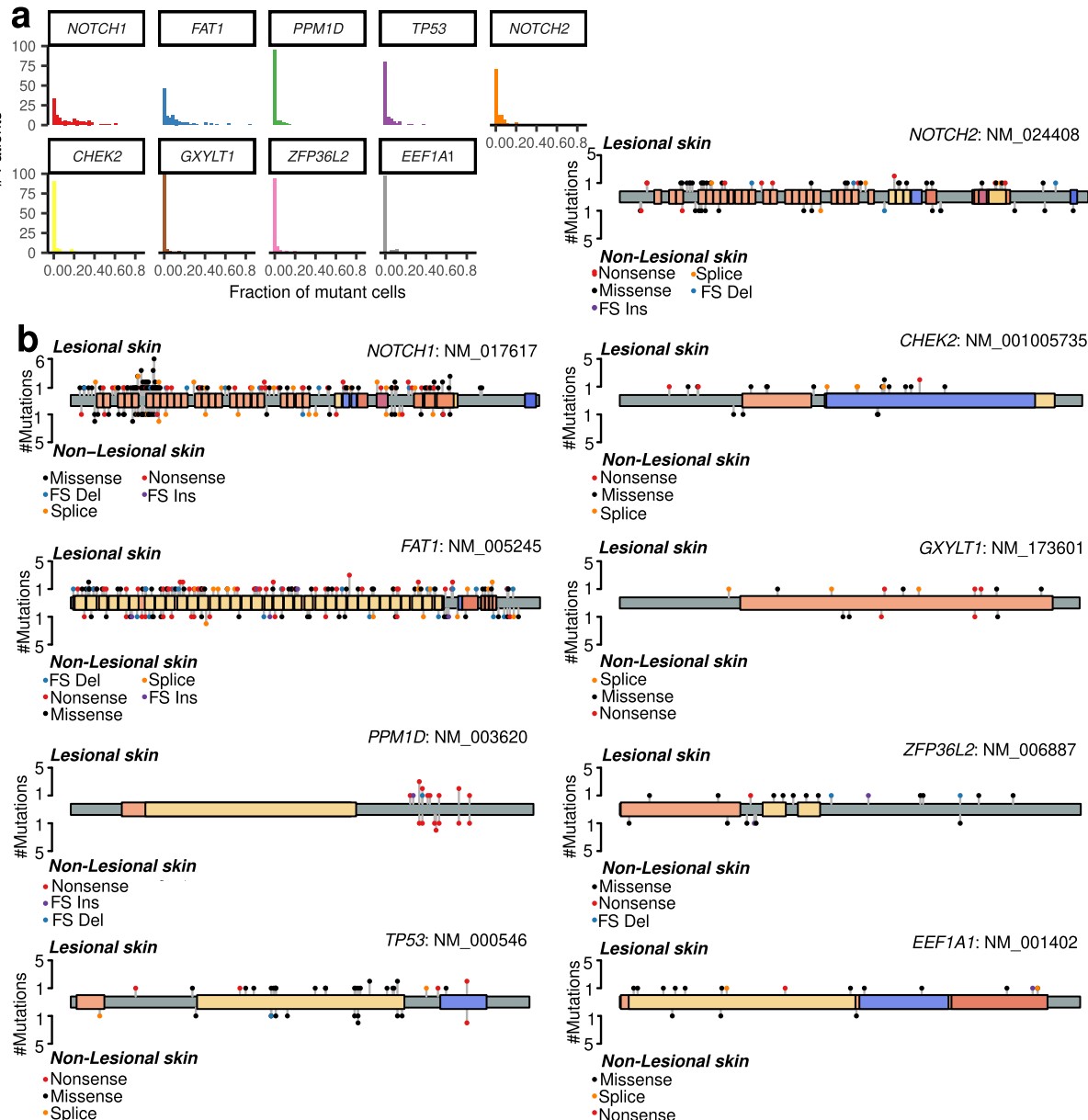

**Extended Data Fig. 5 | Genes showing a significant enrichment of somatic mutations in psoriatic skin. a**) Histogram showing the fraction of cells carrying one or more mutations in each gene for each patient. **b**) Lollipop plots showing the location and annotation of the identified mutations in each gene in lesional (upper) and non-lesional (lower) skin.

# Reporting Summary

## Statistics

For all statistical analyses, confirm that the following items are present in the figure legend, table legend, main text, or Methods section.

| n/a | Confirmed | |
|---|---|---|
| ☐ | ☒ | The exact sample size ($n$) for each experimental group/condition, given as a discrete number and unit of measurement |
| ☐ | ☒ | A statement on whether measurements were taken from distinct samples or whether the same sample was measured repeatedly |
| ☐ | ☒ | The statistical test(s) used AND whether they are one- or two-sided <br> *Only common tests should be described solely by name; describe more complex techniques in the Methods section.* |
| ☐ | ☒ | A description of all covariates tested |
| ☐ | ☒ | A description of any assumptions or corrections, such as tests of normality and adjustment for multiple comparisons |
| ☐ | ☒ | A full description of the statistical parameters including central tendency (e.g. means) or other basic estimates (e.g. regression coefficient) AND variation (e.g. standard deviation) or associated estimates of uncertainty (e.g. confidence intervals) |
| ☐ | ☒ | For null hypothesis testing, the test statistic (e.g. $F$, $t$, $r$) with confidence intervals, effect sizes, degrees of freedom and $P$ value noted <br> *Give P values as exact values whenever suitable.* |
| ☐ | ☒ | For Bayesian analysis, information on the choice of priors and Markov chain Monte Carlo settings |
| ☒ | ☐ | For hierarchical and complex designs, identification of the appropriate level for tests and full reporting of outcomes |
| ☐ | ☒ | Estimates of effect sizes (e.g. Cohen's $d$, Pearson's $r$), indicating how they were calculated |

*Our web collection on statistics for biologists contains articles on many of the points above.*

## Software and code

Policy information about availability of computer code

| | |
|---|---|
| Data collection | No software was used to collect the data. |
| Data analysis | Code supporting the main analyses of the manuscript is provided as R-markdown supplementary code files. Custom scripts documenting the mutation filtering and clustering pipelines, signature extraction and more are publicly available at https://github.com/Solafsson/somaticPsoriasis. <br> The following open source software tools were used as part of our analyses: <br><br> VerifyBamID (v1.1.3) <br> Picard (v.1.131) <br> samtools (v.1.11) <br> CaVEMan (v.1.15.1) <br> cgpPindel (v.3.5.0) <br> Bedtools (v. 2.18) <br> dNdScv (v. 0.0.1.0) <br><br> R-packages: <br> label.switching (v1.8) <br> hdp (v.0.1.5) <br> MutationalPatterns (v. 3.4.0) <br> BSgenome (v.1.60.0) <br> TxDb.Hsapiens.UCSC.hg38.knownGene (v.3.13.0) |

For manuscripts utilizing custom algorithms or software that are central to the research but not yet described in published literature, software must be made available to editors and reviewers. We strongly encourage code deposition in a community repository (e.g. GitHub). See our web collection on Data deposition.

TCGAmutations (v.0.3.0)

For manuscripts utilizing custom algorithms or software that are central to the research but not yet described in published literature, software must be made available to editors and reviewers. We strongly encourage code deposition in a community repository (e.g. GitHub). See the Nature Portfolio guidelines for submitting code & software for further information.

## Data

Policy information about availability of data

All manuscripts must include a data availability statement. This statement should provide the following information, where applicable:
- Accession codes, unique identifiers, or web links for publicly available datasets
- A description of any restrictions on data availability
- For clinical datasets or third party data, please ensure that the statement adheres to our policy

Raw sequencing data are available in the European Genome-phenome Archive (EGA) using study ID EGAS00001004882 and dataset ID EGAD00001011265. Intermediary and supporting files, including mutation calls, mutational cluster assignments, phylogenetic trees, histological images, spatial relationship matrices and more are available in a Mendeley data repository (doi: 10.17632/phvh82vd9g.1). The mutation calls from the TCGA project can be obtained by using the TCGAmutations package in R (https://github.com/PoisonAlien/TCGAmutations). Access to the mutation calls of the HMF cohort can be obtained using the request forms found at https://www.hartwigmedicalfoundation.nl/en/.

## Research involving human participants, their data, or biological material

Policy information about studies with human participants or human data. See also policy information about sex, gender (identity/presentation), and sexual orientation and race, ethnicity and racism.

| | |
|---|---|
| Reporting on sex and gender | We report the sex of all participants in Supplementary Table 1. Data on gender was not collected. Participants were not stratified by sex in any of the analyses reported in the manuscript. |
| Reporting on race, ethnicity, or other socially relevant groupings | All participants are of white-European ancestry. |
| Population characteristics | All participants have been diagnosed with psoriasis vulgaris. The age, sex, disease duration and PUVA treatment history is provided in Supplementary table 1. The age of the participants varied from 18 to 88 and the disease duration varied from 0 to 67 years. |
| Recruitment | We recruited psoriasis patients presenting to the Department of Dermatology, UKSH Kiel. We note that the recruitment is biased in favor of men. The prevalence of psoriasis is thought to be the same in men and women but women only make up 19% of our cohort. It is possible that psoriasis affects men more severely than women, causing them to present to dermatologists more often and/or to be more eager to consent to participate in research into the disease. |
| Ethics oversight | The study was approved by the research ethics committee of Christian-Albrechts University in Kiel (A100/12), the National Health Service (NHS) Research Ethics Committee (Yorkshire & The Humber - South Yorkshire Research Ethics Committee, REC ID 20/YH/0244, IRAS ID 286843) and by the Wellcome Trust Sanger Institute Human Materials and Data Management Committee (approval number 20/0085). |

Note that full information on the approval of the study protocol must also be provided in the manuscript.

## Field-specific reporting

Please select the one below that is the best fit for your research. If you are not sure, read the appropriate sections before making your selection.

☒ Life sciences ☐ Behavioural & social sciences ☐ Ecological, evolutionary & environmental sciences

For a reference copy of the document with all sections, see nature.com/documents/nr-reporting-summary-flat.pdf

## Life sciences study design

All studies must disclose on these points even when the disclosure is negative.

| | |
|---|---|
| Sample size | This is an exploratory study and sample size was not pre-determined. We included 111 patients from whom we had samples available at the time of study initiation. |
| Data exclusions | We excluded from the whole study 6 microbiopsies suspected of being sample swaps and/or contaminated with external DNA. 101 microbiopsies were excluded only from the analysis of structural variants either due to the inability of the algorithm to find an optimal solution or if the goodness-of-fit for the optimal solution was <90%. |
| Replication | The results presented in this manuscript were not replicated in an independent cohort. We did carry out whole-exome sequencing of 18 near-duplicate samples that represent the same histological features on adjacent sections to quantify the sensitivity of our mutation calling pipeline. From this, we estimated a sensitivity of 89% of our mutation calling pipeline as described in the main text. |

| | |
|---|---|
| Randomization | This is a descriptive study with no interventions that recruited only patients and no controls. There was no need for randomization. |
| Blinding | This is a descriptive study with no interventions. The researchers were not blinded to the phenotypes of the participants. |

# Reporting for specific materials, systems and methods

We require information from authors about some types of materials, experimental systems and methods used in many studies. Here, indicate whether each material, system or method listed is relevant to your study. If you are not sure if a list item applies to your research, read the appropriate section before selecting a response.

## Materials & experimental systems

| n/a | Involved in the study |
|---|---|
| ☒ | Antibodies |
| ☒ | Eukaryotic cell lines |
| ☒ | Palaeontology and archaeology |
| ☒ | Animals and other organisms |
| ☒ | Clinical data |
| ☒ | Dual use research of concern |
| ☒ | Plants |

## Methods

| n/a | Involved in the study |
|---|---|
| ☒ | ChIP-seq |
| ☒ | Flow cytometry |
| ☒ | MRI-based neuroimaging |

