## [Peer Review File · Nature Genetics]

Peer Review Information

Manuscript Title: Effects of psoriasis and psoralen exposure on the somatic mutation landscape of the skin.

Corresponding author name(s): Dr Carl (A) Anderson

Reviewer Comments & Decisions:

Decision Letter, initial version:
--

2nd Aug 2022

Dear Dr Anderson,

Your Article entitled "Effects of psoriasis and psoralen exposure on the somatic mutation landscape of the skin" has now been seen by 3 referees, whose comments are attached. In the light of their advice we have decided that we cannot offer to publish your manuscript in Nature Genetics.

While the referees find your work of some interest, they raise concerns that undermine the strength of the novel conclusions that can be drawn at this stage. We feel that these reservations are sufficiently important as to preclude publication of this study in Nature Genetics.

You might want to consider our sister journal *Nature Communications* as a potential venue for the publication of these results. *Nature Communications* publishes high quality and influential research and across the full spectrum of the natural sciences. More information on the journal, the potential benefits of transfer and a link to transfer your paper, can be found at the bottom of this email. Please note that the editorial team at Nature Communications will consider your manuscript independently of our suggestion to transfer.

I haven't consulted with my colleagues at Nature Communications to see if they'd be interested sending a revised manuscript out for review but I'd be happy to do so if it would be helpful to you. Please do let me know if you'd like me to start this process for you.

I am sorry that we cannot be more positive on this occasion but hope that you will find our referees' comments helpful when preparing your paper for submission elsewhere.

With all best wishes,

Safia Danovi
Editor
Nature Genetics

Referee expertise:

Referee #1: somatic mutagenesis

Referee #2: mutational signatures

Referee #3: psoriasis genetics

Reviewers' Comments:

Reviewer #1:

Remarks to the Author:

In their manuscript, Olafsson et al. investigated the clonal structure of the skin affected by psoriasis based on the exome sequencing of 1,182 microbiopsies, including both lesional and non-lesional samples, from 111 psoriasis patients. Major conclusions are:

- #1. Psoriatic skin is highly polyclonal.
- #2. Psoriasis is associated with an increased mutation burden of cell-intrinsic signatures but does not affect clonal expansion substantially.
- #3. PUVA therapy might be associated with larger expansion of clones.
- #4. Psoralen exposure is associated with a unique mutational signature showing T>A, T>C, and T>G at TpA contexts, which are caused by both transcription-coupled repair and damage.
- #4. Cell-intrinsic SBS1/SBS5 are the dominant sources of mutations associated with psoriasis per se and age-related, while the vast majority of mutations are attributed to UV-related SBS7b and 'psoriasis-related' signatures.

Discovery of psoralen-related signature is new and of interest. Otherwise, however, there is a paucity of new findings or conceptual advances to understand the pathogenesis or management of psoriasis. In addition, some of the major conclusions are convincing enough for technical reasons.

Major comments:

1) Estimation of mutation rate

- For estimation of mutation rates, the authors "clustered mutations based on their VAFs to derive a phylogenetic relationship between the clusters relying on the pigeonhole principle" to determine the number of mutations in each clone (Lines 154-160). " However, as the author mentioned, the distribution of median VAFs was skewed to very low values, suggesting the presence of many clones in the majority of samples (Fig. 1d). Thus, in the majority of samples, the pigeonhole principle does not seem to work and therefore, separation of individual clones should be difficult. In such samples, each tip (or leaves) of trees represents clones that were present long before the sampling, which makes the history of clones ambiguous and the following calculation of mutation rates difficult or not valid for correct estimation of mutation rates. The authors are requested to show phylogenetic trees for a number of representative samples.
- Moreover, excluding psoralen-related mutations, the vast majority of mutations in lesional skin samples were UV-related ones, which are known to undergo large variations depending on the site of sampling. The authors estimated the mutation rate excluding psoralen-related mutations to be 14.6 mutations per exome per year. However, this reviewer suspects the validity of this estimation for

above reasons. In fact, mutations numbers are clearly larger for 50-70 years old cases than 70-90 years old cases (Fig. 2b), where the mutation rate does not look age-related.

- The authors stated that the mutation rate was higher in psoralen-exposed skin (Line 314-315), while they also suggested a larger clone size in the psoralen-exposed skin than unexposed one, where the number of mutations are expected to be overestimated. Thus, the higher mutation rate in psoralen-exposed skin might be confounded by the larger clone size, but not determined by psoralen exposure itself.

- As commented on the statements in Lines 118-122 and Lines 154-16, estimation of the mutation rate for SBS1/5 signatures is thought to be very difficult in principle, due to the difference in clone size between lesional and non-lesional skin and an inaccurate estimate of mutation numbers based on phylogenetic tree analysis. According to their methods, reconstruction of phylogenetic trees should have been biased to clones having larger VAF distribution and therefore, the number of mutations was likely to be overestimated (by under-representation of smaller clones). Thus, the estimates of mutation rates in this study should be validated using nanoseq or sequencing of single-cell-derived clones.

2) Mutation signature

- Lines 201-208: In PUVA-treated samples, the vast majority of mutations were either SBS7b or psoralen-related ones. This reviewer does not believe it appropriate to estimate the rate of intrinsic mutations, i.e., SBS1/SBS5, or to separate SBS1 and SBS5 signatures, by simply subtracting the number of UV- and psoralen-related mutations. Influence of the large number of UV- and PUVA-related mutations looks inevitable; it may also influence the number of mutations assigned to other signatures. For example, mutations numbers are clearly larger for 50-70 years old cases than 70-90 years old cases.

- The authors correlate the elevated cancer incidence associated with psoriasis with PUVA therapy rather than psoriasis per se (Lines 442-444), which is in line with the finding of epidemiological studies. Given that, one expects an increased psoralen-burden of psoralen-related mutations in skin cancers. The authors can test this using for example TCGA data. Moreover, the authors stated that PUVA-treatment causes a large number of mutations having a unique signature without resulting in malignant transformation. This is confusing.

3) Comparison of clone size between lesional and non-lesional skin

- The authors conclude that psoriasis disease may not substantially influence clonal expansion based on the equivalent medians of median VAFs of mutations in individual lesional and non-lesional samples. However, this may not support their claim. For example, when cumulative VAFs of all mutations in individual lesional samples (not just individual medians) are evaluated in both groups, lesional samples may have a larger fraction of mutations showing large VAF values, which should suggest a positive effect of psoriasis on clone size. Comparison of median VAFs of individual samples alone may not correctly evaluate expansion of clones (Lines 118-122).

- Related to 1), on the other hand, the authors suggest that a high burden of the psoralen signature is associated with clonal expansions of keratinocytes in their dataset based on the analysis of shared mutations in paired samples correlated with the distance between them (Lines 314-333). Can the authors also assess the clone size of psoralen-exposed skin based on compare the median VAF size between samples with high burden of the psoralen signature and those with no psoralen signature as shown in Fig. 1d?

4) Detection of positively selected mutations (Line 372)

All these samples had thousands of mutations in exome sequencing, suggesting the presence of high noises, which might have prevented sensitive detection of positively selected mutations. Analysis of increasing number of samples might have enable detection of new drivers. In fact, in the case of the analysis of liver samples by the same authors, ~1000 samples were required to identify positively

selected mutations, which had been not detected with 300-400 samples. The number of background mutations in the skin are much larger than that in the liver, suggesting a compromised sensitivity to detect positively selected mutations.

Minor issues:

- 5) Psoriatic plaques repeat expansion and shrinkage throughout life. Thus, non-lesional skin at the time of sampling may experience repeated inflammation and may not necessarily work as a good control. Moreover, the mutation rate therein might be overestimated because of the past exposure to inflammatory processes. In addition, lesional skin also have such history of repeated inflammation and remission.
- 6) While the median size of microbiopsies is $<0.01\text{mm}^2$, it would be helpful to provide an estimate of an approximate number of cells in typical microbiopsies as well.
- 7) While the median size of microbiopsies is $<0.01\text{mm}^2$, it would be helpful to provide an estimate of an approximate number of cells in typical microbiopsies as well.
- 8) Line 181: Patient 34 was described as an outlier. However, Fig.2A or Suppl. Fig. 3 has no Patient IDs, which makes identification of the case (Patient 34) impossible.
- 9) Fig. 2b: The outlier case (Patient 34) should be excluded from the regression analysis.
- 10) In the discussion in Lines 209-224, the result from two seminal papers (#24, #25) in IBD should be also cited and discussed.
- 11) Lines 334-338: "Compared with a uniform mutational process, the psoralen-signature is 1.37 times more likely to result in splicing mutations in genes reported to be recurrently mutated in squamous cell carcinomas and normal skin. It would be especially likely to disrupt AT-AC introns, which have 5'-AT and AC-3' boundaries, in contrast to the typical 5'-GT and AG-3' splice sites." No data are shown to support these statements.
- 12) Line 484-486: EEF1A1 is described as being identified as a driver gene in urothelium. However, this may not support the statement "EEF1A1 mutations are generally selected for in squamous epithelia, because urothelium is not squamous cell epithelium but is typical transitional epithelium.
- 13) What is the criteria for "extensive" PUVA treatment (Line 533)? Usually, >200 sessions are commonly used to define very frequency therapy in epidemiological studies. Also, better define "limited number of PUVA" (Page 23, line 538). In Xpa deficient mice, only 1 UVB exposure is sufficient to cause skin cancer in a few months.
- 14) Many typos and errors
 Page 2, line 62: "biopsie" => "biopsy", "patients"=> "patients"
 Page 4, line 1: "DNA extracted from each of the microbiopsies was whole-exome sequencing": some words missing?
 Page 16, Fig. 3: characters are barely readable in fig 3b and c.
 Line 395: "failure detect" => "failure to detect"
 Page 19, Fig. 4 legend: "number" rather than "umber"
 Page 21 "Wehypothesize" => "We hypothesize"

Reviewer #2:

Remarks to the Author:

NG-A60455

Effects of psoriasis and psolaren exposure on the somatic mutation landscape of the skin

Summary

Psoriasis is a complex autoimmune disease, which often manifests as chronic inflammation of the skin. In this article, 1182 skin microbiopsies from 111 patients with psoriasis were whole exome sequenced to characterise the somatic mutations affecting psoriatic skin, in contrast with normal skin. The results indicate that there is no significant difference between the clonality composition or driver selection of normal skin and psoriatic skin. There is a difference in mutational burden of only specific mutational processes, namely those related to SBS1 and SBS5, the age-associated mutational signatures. More specifically, psoriatic skin appears to have increased mutation accumulation from these mutational processes only.

The study then moves on to characterise the mutational signature of psoralens exposure, which are chemicals used along with UV-A (PUVA) to treat psoriasis. Psoralens' mutagenic effects seem to be mostly single nucleotide variants of T>N at TpA sites. Additional characterisation of this signature helps to understand the mechanistic causes of the mutations. The authors postulate that the moderate increase in skin cancer risk for people affected by psoriasis might be due in part or fully by the PUVA treatment.

The authors provide interesting comparisons between the findings in this paper and what is already known about the somatic landscape and cellular evolution of the normal skin, as well as of a related autoimmune disease, the inflammatory bowel disease.

Overall, the article is well written, the data is of high interest and the analysis is of high standard, with appropriate evidence for the authors' claims. I have found only some minor issues that are mostly related to some of the figures, see below.

Minor issues:

1. I think that more explanation is required to understand the linear model described in section 7.2 of methods. Currently, there is only a high level explanation and a supplementary R code. Neither the R code nor the current methods provide a detailed description of the mathematical model and its assumptions. Please provide these details.
2. I might have missed it, but I didn't find a mention to the possibility that SBS1/5 might increase in psoriasis also because the psoriasis cells undergo hyperproliferation, so more cell cycles and perhaps more chances to introduce SBS1/5 mutations. Please add a few lines in the discussion to mention this possibility.
3. I find the histograms in Figure 1d and Supplementary Figure 1a-c confusing and I would like them to be improved. The way they are currently it is hard to tell which bar corresponds to which interval, because the x axis is continuous. This type of histogram would work for categorical data with each category labelled, but I find it unclear here. Perhaps just draw the two histograms separately.
4. Figures 2b-d also need improvement for clarity. The main issues I find here are that the blue dots are hard to see, mostly covered by red dots. Also there is no indication of what the line and shaded region around the line are, this information should be added. Also, given the question of whether lesional and non-lesional skin are different, we should probably look at two slopes here, one for lesional and one for non-lesional. I suggest drawing lesional and non-lesional data separately and reporting some slope statistics on the figure.
5. At line 334 and following, some interesting findings about splicing mutations are reported, however

there seems to be an indication of where to find these results. Is there a table with the associated values? Is there a supplementary file with the procedure? This is also missing from the methods section. Please add.

6. The text in Figure 3, supplementary figure 5, supplementary figure 6 some text is too small, please improve readability.

7. Figure 3f, Psoralen misspelt Poralen

Reviewer #3:

Remarks to the Author:

Comments on "Effects of psoriasis and psoralen exposure on the somatic mutation landscape of the skin" by Dr. Olafsson and colleagues. In this study the authors performed whole-exome sequencing of 1182 microbiopsies dissected from lesional and non-lesional epidermis from 111 patients with psoriasis to search for evidence that somatic mutations in keratinocytes may influence the disease process and to characterize the effects of the disease on the mutation landscape of the epidermis.

This study is similar to what this group has previously done on colonic mucosa in inflammatory bowel disease, published in 2020 in Cell (PMID:32697969) making the contrast with psoriasis more notable, given the very marked differences in findings between these two inflammatory diseases. There are several novel findings reported here that are of importance to the dermatology/medical field. This includes the findings that psoriasis is associated with increased mutation burden of the cell-intrinsic signatures SBS1 and SBS5 but not of UV-light. Further, the finding that despite keratinocyte hyperproliferation lesional psoriatic skin remains highly polyclonal showing no evidence of spread of clones carrying potentially pathogenic mutations, is highly interesting. Lastly, the authors detect distinct PUVA-associated mutational signature in the genomes of exposed cells that is tightly linked with transcription.

I agree with the authors these results suggest that somatic mutations in keratinocytes are unlikely to influence the pathogenesis of psoriasis.

I have the following comments.

It is well established that PUVA treatment is carcinogenic in patients that have received multiple treatments. For this reason, the use of PUVA has markedly diminished and is currently rarely used, if at all, for the treatment of psoriasis. Therefore, the emphasis on this aspect of the study should be toned down, although I agree that it is very interesting. In addition, I'd expect that most of the patients that had history of PUVA treatment were older than non-PUVA treated individuals and I think the authors should address that and how the age difference in PUVA + vs PUVA- subjects, would potentially confound the data/results and their interpretation. Also, the authors calculate mutation burden of keratinocytes clones per year, but it would be of interest if the authors can show this separately for the PUVA treated group. (and not include that in the initial calculation). PUVA carcinogenesis is associated with the number of PUVA treatments, did the authors compare that in their study cohort?

I'm surprised that the authors do not address potential mutation burden and analyses for patients treated with UVB, which is the most commonly used phototherapy used to treat psoriasis patients, and it's surprising that this is not addressed here at all. Also, from reading the manuscript it is not clear

how, and if, the authors distinguish natural UV exposure from UV-treatment related exposure. In addition, and related to that, do the authors see site-specific differences in their UV signatures? i.e., chronically sun exposed sites (i.e., legs/arms) vs. more sun-protected sites (flank/abdomen sites)?

What about other treatments? Many of these patients may have been treated with methotrexate and other antimetabolite treatments that will directly interfere with DNA synthesis. This should be addressed by the authors as well.

Line 506. I'm not sure that the argument that both IBD and psoriasis are Th17 mediated chronic inflammatory diseases as the role of IL-17 in IBD seems to be very different, with anti-IL-17 drugs sometimes unmasking or triggering IBD, while highly effective in treating psoriasis, suggesting a completely different role of IL-17A in those two diseases.

Line 541. I'm not aware of psoralens being still used in sunscreens as this practice has been banned in most countries more than 20 years ago. Therefore, this argument has low relevance and should be removed. I agree with the authors about the potential in regards to consumption of furocoumarin-rich foods and this should be kept although this in itself is not a focus of the current manuscript and I do not think the authors have sufficient data on this particular aspect of furocoumarins to make any definite statements on that .

Patient demographics are incomplete and need to be expanded – also the format as listed is not easily approachable and should be revised in a table format. Treatment only includes PUVA but should also include other treatments, in particular UVB and MTX as those are both potential mutation inducing agents.

Minor comment

Line 464....lacks space between two words "Wehypothesize"

Decision Letter, Appeal

18th Jan 2023

Dear Carl,

Thank you for your message of 18th Jan 2023, asking us to reconsider our decision on your manuscript "Effects of psoriasis and psoralen exposure on the somatic mutation landscape of the skin". I have now discussed the points of your appeal with my colleagues, and we would be happy to send the revised manuscript back out to the original reviewers.

Owing to a quirk in the way we process appeals, we now need to you re-upload all your files. I appreciate that this is highly annoying and I apologise! When preparing a revision, please ensure that it fully complies with our editorial requirements for format and style; details can be found in the Guide to Authors on our website (<http://www.nature.com/ng/>).

Please be sure that your manuscript is accompanied by a separate letter detailing the changes you have made and your response to the points raised. At this stage we will need you to upload:

- 1) a copy of the manuscript in MS Word .docx format.
- 2) The Editorial Policy Checklist:
<https://www.nature.com/documents/nr-editorial-policy-checklist.pdf>
- 3) The Reporting Summary:
<https://www.nature.com/documents/nr-reporting-summary.pdf>
(Here you can read about the role of the Reporting Summary in reproducible science:
<https://www.nature.com/news/announcement-towards-greater-reproducibility-for-life-sciences-research-in-nature-1.22062>)

Please use the link below to be taken directly to the site and view and revise your manuscript:

[redacted]

Kind regards,

Safia Danovi
Editor
Nature Genetics

Author Rebuttal to Initial comments

Somatic mutations in psoriasis - rebuttal

Referee expertise:

Referee #1: somatic mutagenesis

Referee #2: mutational signatures

Referee #3: psoriasis genetics

Reviewers' Comments:

Reviewer #1:

Remarks to the Author:

In their manuscript, Olafsson et al. investigated the clonal structure of the skin affected by psoriasis based on the exome sequencing of 1,182 microbiopsies, including both lesional and non-lesional samples, from 111 psoriasis patients. Major conclusions are:

#1. Psoriatic skin is highly polyclonal.

#2. Psoriasis is associated with an increased mutation burden of cell-intrinsic signatures but does not affect clonal expansion substantially.

#3. PUVA therapy might be associated with larger expansion of clones.

#4. Psoralen exposure is associated with a unique mutational signature showing T>A, T>C, and T>G at TpA contexts, which are caused by both transcription-coupled repair and damage.

#4. Cell-intrinsic SBS1/SBS5 are the dominant sources of mutations associated with psoriasis per se and age-related, while the vast majority of mutations are attributed to UV-related SBS7b and 'psoriasis-related' signatures.

Discovery of psoralen-related signature is new and of interest. Otherwise, however, there is a paucity of new findings or conceptual advances to understand the pathogenesis or management of psoriasis.

In addition, some of the major conclusions are convincing enough for technical reasons.

There is growing interest in understanding the potential roles of somatic mutations in non-neoplastic diseases. However, the role of somatic mutation in most of these diseases remains completely unexplored. Among the few such diseases which have been studied, there are some results which are potentially transformative for our understanding of the disease process. One such disease is IBD, where we and others have shown that inflammation is associated with clonal expansions in the gut and that mutations in some genes are under positive selection in intestinal epithelial cells, including genes in the IL17 pathway.

Understanding how somatic mutations occur and spread in other inflammatory diseases could similarly help us understand the progression of those diseases and to understand the similarities and differences that exist between related diseases affecting different tissues.

Given the known and important role of IL-17 in psoriasis, a chronic inflammatory disease that like IBD is characterised by inflammation of epithelial tissue, we feel that the questions of whether selective pressure and clonal dynamics are changed in psoriasis-epidermis are both highly relevant and exciting. We have answered those questions and show that psoriasis has a minimal effect on the selection and spread of keratinocyte clones within the epidermis, thus providing novel insights into the biology of psoriasis. Our result also adds complexity to the role of somatic mutations in common complex disease, because we now know that even among inflammatory diseases of the epithelia the role of somatic

mutation varies. Our paper demonstrates that further research into the different roles of somatic mutations in inflammatory diseases is needed.

We disagree with the reviewer that technical reasons inhibit our ability to draw conclusions from our data. Below, we provide a more detailed response to the technical concerns raised by the reviewer, outlining the strength of evidence supporting our main conclusions.

Major comments:

1) Estimation of mutation rate

- For estimation of mutation rates, the authors “clustered mutations based on their VAFs to derive a phylogenetic relationship between the clusters relying on the pigeonhole principle” to determine the number of mutations in each clone (Lines 154-160).” However, as the author mentioned, the distribution of median VAFs was skewed to very low values, suggesting the presence of many clones in the majority of samples (Fig. 1d). Thus, in the majority of samples, the pigeonhole principle does not seem to work and therefore, separation of individual clones should be difficult. In such samples, each tip (or leaves) of trees represents clones that were present long before the sampling, which makes the history of clones ambiguous and the following calculation of mutation rates difficult or not valid for correct estimation of mutation rates. The authors are requested to show phylogenetic trees for a number of representative samples.

We agree with the reviewer that estimating the mutation burden from microdissected bulk cells can be challenging. Indeed, this is one of the major limitations of using the LCM approach in a polyclonal tissue such as the skin. However, the median VAFs of lesional and non-lesional skin are 0.25 and 0.26, respectively. This is similar to the median VAFs reported in many cancer sequencing studies and we thus do not believe the distribution of VAFs is “skewed to very low values”. A tail of low-VAF mutations is missed in all studies, cancer or normal tissue, that do not use some variant of single-cell or single-molecule sequencing. Although the mutation rate we have estimated is subject to some uncertainty owing to low VAFs in a small fraction of the microbiopsies (just as in cancer sequencing studies), it is nevertheless the best estimate in the literature. Past studies of normal skin have used targeted sequencing panels and sequenced biopsies of skin two orders of magnitude larger than those in the current study. The samples have thus been a mixture of very many clones and, unfortunately, these studies have not been able to assess the mutation burden of the skin at all. ***We are thus the first to provide an estimate of the mutation rate of the skin in both health and psoriasis.***

While it is true that the low VAFs in a minority of biopsies add some uncertainty to the phylogenetic tree building for some individuals, we have two advantages over traditional cancer sequencing studies which help overcome the problem.

Firstly, we have sequenced multiple microbiopsies which are close together in space. When a clone stretches over multiple microbiopsies, this helps resolve ambiguities. For example, we have added the figure below as a panel to Figure 1: Consider the purple clone (number 12) and the gray clone (number 4) in microbiopsy F. Clones 12 and 4 are found at similar VAFs in microbiopsy F. However, the VAF of clone 12 is greater in microbiopsy E and in microbiopsy G, the gray clone 4 is not seen at all. This supports the notion that clone 12 is ancestral to clone 4. A second example is the beige clone 3 and the green clone 1. Their combined VAFs in microbiopsy A exceed 1, so the pigeonhole principle clearly indicates that one is a subclone of the other. We can combine that with evidence from microbiopsy B to show that clone 1 is clearly ancestral to clone 3.

Secondly, the very small size of the microbiopsies (typically <100um by <60um) means that there is very little room for multiple clones to co-exist and yet for each to be prevalent enough to pass the detection limit of this study. As a consequence, most of the trees have a simple structure and it is common for only a single clone, or for a clone and its subclone, to pass the detection limit in a particular microbiopsy (for example microbiopsies C, D and E below). This is an important reason why, from 1182 microbiopsies, we identify only 1196 clones (tips of the phylogenetic trees). Even when only a single clone passes the detection limit for a microbiopsy, that clone will still have $VAF < 0.5$ as the microbiopsy from which it derives is “contaminated” both by tissue infiltrating immune cells and by a mixture of cells from different neighbouring clones, none of which may be present at a high enough frequency to pass the detection threshold.

For the randomly chosen patient below for example, there is little reason to doubt the validity of the pigeonhole principle. While we recognize that across the dataset as a whole, there will be instances where there is uncertainty regarding whether a cluster represents a subclone of another clone in the biopsy or an independent clone, we feel this adds uncertainty to our mutation burden estimate but does not invalidate it.

Figure 1F: (Top) Histology image of a lesional biopsy (haematoxylin and eosin staining). Examples of microbiopsies of epidermis are highlighted. The scale bar shows 250 micrometers. (Middle) The estimated cellular fractions of the mutation clusters with VAF > 1% found within the microbiopsies above. (Bottom) a phylogenetic tree for the patient based on the mutation clusters above. The rightmost branch is from a non-lesional biopsy not shown.

Furthermore, the clonal structure of the epidermis appears unchanged in lesional compared with non-lesional skin (the median VAFs are 0.25 and 0.26, respectively, and few instances of large-scale clonal spread are observed in the phylogenetic trees). Thus, the limitations of the LCM approach will affect the lesional and non-lesional samples equally and so would not drive a difference in the mutation burden between groups.

The figure above is now a part of Figure 1 in the main text of the revised manuscript. Phylogenetic trees for all donors in the study and cellular fraction estimates of all mutation clusters are provided in a Mendeley data repository under a doi specified in the Data and materials availability statement.

<https://data.mendeley.com/datasets/rfcy88sb9s>

- Moreover, excluding psoralen-related mutations, the vast majority of mutations in lesional skin samples were UV-related ones, which are known to undergo large variations depending on the site of sampling. The authors estimated the mutation rate excluding psoralen-related mutations to be 14.6 mutations per exome per year. However, this reviewer suspects the validity of this estimation for above reasons. In fact, mutations numbers are clearly larger for 50-70 years old cases than 70-90 years old cases (Fig. 2b), where the mutation rate does not look age-related.

Excluding psoralen-related mutations, there is a strong linear relationship with age in our dataset. The seemingly lower-than-expected mutation burdens for the four 70-90 year old cases in Fig. 2c (b in the figure below), which are not statistically significantly lower than expected, stems from the fact that these patients happen to have donated biopsies from anatomical sites which are not highly sun-exposed. Two of the patients in this age range donated biopsies from the thigh, one from the flank and one from a buttock. Fig 2d (Fig 2c in the revised version) shows the mutation burden of SBS1/5 and there these individuals have mutation burdens within the expected range for their age. In our modeling of the mutation burden, we included a fixed effect for the anatomical site of the biopsy in order to capture the different mutation burdens due to varying UV-exposure of different anatomical locations.

The revised figures for the total mutation burden and the SBS1/5 specific mutation burden are below:

- The authors stated that the mutation rate was higher in psoralen-exposed skin (Line 314-315), while they also suggested a larger clone size in the psoralen-exposed skin than unexposed one, where the number of mutations are expected to be overestimated. Thus, the higher mutation rate in psoralen-exposed skin might be confounded by the larger clone size, but not determined by psoralen exposure itself.

We do not state that the mutation burden of signatures other than the psoralen signature is different between psoralen exposed and never exposed skin. We do state that some samples show a unique mutational signature which we attribute to psoralen exposure. However, we estimate the number of mutations attributed to this signature and then subtract them from the total mutation burden before fitting our linear models of mutation burden against age. Thus we don't include psoralen-related mutations in our linear models of the mutation burden and they do not contribute to our estimate of 14.6 mutations per exome per year. We have made minor changes to the text and figure legends to clarify this.

We note that "psoralen-exposed" is not synonymous with "lesional". The patients who have a history of PUVA treatment have mostly received this in a UV-chamber, exposing most of their bodies to the treatment. The psoralen signature is seen also in non-lesional samples and indeed, the clone showing the highest burden of the signature is from non-lesional skin. Increased clonality as a result of psoralen

exposure would therefore not affect the comparison of the mutation rate between lesional and non-lesional samples.

- As commented on the statements in Lines 118-122 and Lines 154-16, estimation of the mutation rate for SBS1/5 signatures is thought to be very difficult in principle, due to the difference in clone size between lesional and non-lesional skin and an inaccurate estimate of mutation numbers based on phylogenetic tree analysis. According to their methods, reconstruction of phylogenetic trees should have been biased to clones having larger VAF distribution and therefore, the number of mutations was likely to be overestimated (by under-representation of smaller clones). Thus, the estimates of mutation rates in this study should be validated using nanoseq or sequencing of single-cell-derived clones.

As better addressed in our reply to the first comment of this reviewer, we acknowledge that there are limitations inherent to the LCM approach that make the estimation of the mutation rate more difficult. The estimation of the absolute mutation rate in the skin is subject to these limitations but is also a minor part of our results. More important than the absolute rate is the relative difference in SBS1/5 burden between lesional and non-lesional skin. The reviewer claims that the differences in clonality between lesional and non-lesional skin would make the comparison of mutation burden difficult. However, as we state in both the abstract and the first section of the results, there are no differences in the clonal structure between lesional and non-lesional skin (they have median VAFs of 0.25 and 0.26, respectively) and so this would not bias the estimate of SBS1/5 burden. The limitations of the LCM approach, which add uncertainty to the absolute mutation burden estimate, would be expected to affect lesional and non-lesional samples equally and not affect the estimate of the relative differences between the two skin types.

Furthermore, we have taken steps to limit the effects of VAF and coverage on our mutation burden estimate. The length of each branch of the phylogenetic trees is adjusted based on a sensitivity estimate calculated from the coverage and the VAF of the mutation cluster which forms the branch. The approach is described in detail in section 7.1 of the methods section.

2) Mutation signature

- Lines 201-208: In PUVA-treated samples, the vast majority of mutations were either SBS7b or psoralen-related ones. This reviewer does not believe it appropriate to estimate the rate of intrinsic mutations, i.e., SBS1/SBS5, or to separate SBS1 and SBS5 signatures, by simply subtracting the number of UV- and psoralen-related mutations. Influence of the large number of UV- and PUVA-

related mutations looks inevitable; it may also influence the number of mutations assigned to other signatures. For example, mutations numbers are clearly larger for 50-70 years old cases than 70-90 years old cases.

Before modeling the mutation rate, we first extracted mutational signatures for all tips of the phylogenetic trees (referred to as clones). The mutational signature of psoralen exposure does not correlate with age and causes some clones to be outliers (Fig 2b in the initial submission, that panel has been removed in the revised version). It therefore makes sense to subtract the number of mutations assigned to this signature if one wishes to model how mutations attributed to other mutagenic processes accumulate through time.

Contrary to the reviewer's comment, we do not estimate either the mutation burden attributed to UV-light or the mutation burden attributed to SBS1/5 by simple subtraction. We agree with the reviewer that this would not be appropriate. Instead, as described in lines 1050-1053 of the methods section, we use a hierarchical Dirichlet process to estimate the burden of each individual signature in each clone separately. The estimated burden of each individual signature is used as a response variable in our linear-mixed effects modeling.

The only subtraction done is when Psoralen-associated mutations are removed from the total before regressing the mutation burden against the age of the patient. This is appropriate because the psoralen mutations do not accumulate with age, but rather we suppose they are found in proportion with the extent of psoralen+light exposure. We have made minor changes to the text to clarify this point.

That mutation numbers seem larger for 50-70 year olds than 70-90 year olds in Fig 2c can likely be explained by the anatomical location of the biopsies. The biopsies from the 70-90 year olds come from the buttock or the thigh, and so can be supposed to have not been heavily sun-exposed during the lifetime of the individuals. Indeed, the burden of mutations attributed to SBS1/5 in these individuals (Fig 2d, Fig 2c in the revised version) follows the age line more closely. The effect of anatomical sites is already accounted for in our model, although the effects of covariates are not plotted in figures 2b-d.

- The authors correlate the elevated cancer incidence associated with psoriasis with PUVA therapy rather than psoriasis per se (Lines 442-444), which is in line with the finding of epidemiological studies. Given that, one expects an increased psoralen-burden of psoralen-related mutations in skin cancers. The authors can test this using for example TCGA data. Moreover, the authors stated that

PUVA-treatment causes a large number of mutations having a unique signature without resulting in malignant transformation. This is confusing.

We agree with the reviewer that one hypothesis that comes from our work is that skin cancers may show an increased burden of psoralen-related mutations. Following the reviewer's suggestion, we looked for evidence of the psoralen mutational signature in data generated from cutaneous melanoma samples by the TCGA, but failed to find any evidence of it. This tells us that psoralen exposure does not make a major contribution to sporadic skin cancers at the population level. We were not surprised by this finding because psoralens are not commonly prescribed and used. For example, at most 2% of the world's population have psoriasis and only a minority of those will be prescribed psoralens.

Although the signature is not found in TCGA, since the preprint has become available, we have been contacted by Axel Rosendahl Huber at the Institute for Research in Biomedicine in Barcelona, who has found the signature to contribute a large number of mutations in a single basal-cell carcinoma sample in the Hartwig Medical Foundation dataset. Upon review of his medical records, this patient was found to have a history of psoriasis and of PUVA treatment. This tells us that while psoralen exposure makes little contribution to skin sporadic skin cancer risk at the population level, it can have profound consequences on skin cancer risk at the individual level. Recognizing Axel's contribution to the study, we have added him to the author list and made the following changes to the manuscript:

1. We have added the mutational spectrum of the exposed sample from the Hartwig Medical Foundation to Figure 3. Panel 3a now compares the mutational spectra of one heavily exposed microbiopsy from our cohort and a cancer case from the HMF:

2. We have added the following paragraph to the results section where we discuss the psoralen signature:

“We did not find evidence of the psoralen signature in any of 468 cases of cutaneous melanoma samples in the cancer genome atlas (TCGA) (Cancer Genome Atlas Network 2015). However, one patient with basal cell carcinoma in the Hartwig Medical Foundation (HMF) cohort (Priestley et al. 2019), HMF003357A, did show clear evidence of the signature (Figure 3a). A HMF study coordinator reviewed this patient’s medical records and found that he has a history of psoriasis and PUVA treatment. This provides an orthogonal validation of the psoralen signature in an independent cohort and suggests that while psoralen exposure is a rare source of mutagenesis in sporadic skin cancers, it can come to dominate the mutational spectrum in patients who have received PUVA treatment, even drowning out the effects of UV-light. “

3. We have added the following paragraph to the discussion, where we discuss other potential sources of psoralen mutations than PUVA:

“While psoralen exposure in the context of psoriasis is particularly likely to occur during PUVA-therapy, other means of exposure include the past use of sunscreen or tanning lotions containing psoralens or even the consumption of furocoumarin-rich foods. It has been hypothesised that orally ingested furocoumarins increase risk of skin cancers⁵⁶⁻⁵⁹. We looked for evidence of the psoralen signature both in TCGA melanoma samples and in skin cancers from the HMF and found evidence of the signature only in a psoriasis patient with a history of PUVA treatment. This does not rule out a mutagenic effect of furocoumarins from diet, but suggests that they are rarely a major source of mutations in the populations from which these cohorts are drawn. We propose that testing for a relationship between furocoumarin consumption and the psoralen mutational signature in sun exposed skin offers a mechanistic way to test the hypothesis that furocoumarin rich foods cause skin cancers.”

Regarding the confusion of the reviewer on the effect of PUVA on cancer risk: In the final paragraph of the manuscript, we stated that:

“PUVA treatment, and likely psoralen exposure in general, leaves a distinct mutational signature on the genomes of exposed cells and can result in a very large number of somatic mutations without resulting in malignant transformation.”

The purpose of this statement is not to claim that PUVA treatment can't result in cancer or doesn't increase the risk of cancer. It almost certainly does. Instead, the statement is to highlight that in the paper, we provide an example of a patient with a history of PUVA treatment and over 160,000 substitutions have accumulated in a clonal sample of normal skin from this individual without that cell undergoing malignant transformation. This adds to the growing body of evidence that normal cells can suffer a very large number of somatic mutations without undergoing malignant transformation.

We have changed the statement so it now reads:

“PUVA treatment, and likely psoralen exposure in general, leaves a distinct mutational signature on the genomes of exposed cells and can result in a very large number of somatic mutations. The increased mutation burden almost certainly serves to increase the risk of malignancy, although in our whole-genome sequencing data we also found an example of a clone of skin cells which has accumulated over 160,000 somatic mutations without this resulting in malignant transformation. “

3) Comparison of clone size between lesional and non-lesional skin

- The authors conclude that psoriasis disease may not substantially influence clonal expansion based on the equivalent medians of median VAFs of mutations in individual lesional and non-lesional samples. However, this may not support their claim. For example, when cumulative VAFs of all mutations in individual lesional samples (not just individual medians) are evaluated in both groups, lesional samples may have a larger fraction of mutations showing large VAF values, which should suggest a positive effect of psoriasis on clone size. Comparison of median VAFs of individual samples alone may not correctly evaluate expansion of clones (Lines 118-122).

- Related to 1), on the other hand, the authors suggest that a high burden of the psoralen signature is associated with clonal expansions of keratinocytes in their dataset based on the analysis of shared mutations in paired samples correlated with the distance between them (Lines 314-333). Can the authors also assess the clone size of psoralen-exposed skin based on comparing the median VAF size between samples with high burden of the psoralen signature and those with no psoralen signature as shown in Fig. 1d?

With regards to the first point, we are principally interested in discovering large-scale differences in the clonal structure of the skin. In both IBD and liver disease, clone sizes are *qualitatively* different between the normal tissue and disease, with clones spreading over millimeters or even centimeters of tissue. Comparing the VAF distributions of lesional and non-lesional skin is one way of showing that large-scale clonal expansions are not observed in psoriasis but this conclusion is supported by other types of comparison as well. In the revised version of the manuscript, panels c-e of Figure 1 now look as shown below. Figure 1e shows an anecdotal example of a randomly chosen lesional biopsy. The reader can

appreciate from this that, for this individual, the mutational clusters do not extend across large areas, although the green and the purple clusters are seen in more than one biopsy. This is typical for the dataset and is probably the strongest evidence for lack of large-scale clonal expansions. Figure 1d shows the fraction of mutations that are shared between pairs of microbiopsies as a function of the distance between them for all microbiopsy pairs in the dataset. We do not observe any differences in the fraction of mutations shared between lesional and non-lesional skin biopsies.

Finally, Figure 1c shows the median VAFs of the microbiopsies from lesional and non-lesional skin. These have identical medians but the reviewer hypothesizes that lesional samples may have a greater fraction of high-VAF mutations. Below, we compare the VAF-quantiles between lesional and non-lesional samples. The plot shows that all distributions, including the one showing the 75% VAF quantile, are highly similar, suggesting no systemic shift to high-VAF mutations in the lesional samples. In particular, the distributions of the 75% VAF quantiles are not significantly different ($p=0.22$, Kolmogorov-Smirnov test) as would be expected if lesional samples had a larger fraction of mutations with large VAFs as suggested by the reviewer.

All these lines of evidence make our conclusion that clonal structure is unchanged in lesional compared with non-lesional skin in psoriasis patients highly likely to be true. If the clone sizes are different, it will be a matter of a few micrometers, a difference that would be of very uncertain biological relevance. Imagine the median size of a keratinocyte clone was shifted from a diameter of 50 microns to 55 or 60 microns. The effects this would have on the biological interpretation of the paper would be very modest.

With regards to the second point, the reviewer seems to be requesting the analysis which we previously presented in Figure 3e. In this figure, we compared the median VAFs between samples with evidence of psoralen exposure and samples without evident exposure. Another reviewer pointed out that this analysis did not take the age of the patients into account and that this was likely to influence the analysis as patients who have received PUVA treatment are likely older than those who have not. In response to this, we have amended our analysis and replaced Figure 3e with the following panel, which compares the median VAFs across all microbiopsies from a patient as a function of the patient's age. The conclusion is the same, patients showing evidence of the psoralen signature have a significantly higher median VAFs than those who do not after correcting for age.

4) Detection of positively selected mutations (Line 372)

All these samples had thousands of mutations in exome sequencing, suggesting the presence of high noises, which might have prevented sensitive detection of positively selected mutations. Analysis of increasing number of samples might have enable detection of new drivers. In fact, in the case of the analysis of lever samples by the same authors, ~1000 samples were required to identify positively selected mutations, which had been not detected with 300-400 samples. The number of background mutations in the skin are much larger than that in the liver, suggesting a compromised sensitivity to detect positively selected mutations.

It is not true that all the samples had thousands of mutations in the exome sequencing. The median number of mutations detected per clone is 357. Only 17% of the clones have mutation burdens >1000. Owing to the reviewer's use of the word "noise", we would like to highlight that we have done repeated sequencing of the same histological features and observed 89% of mutations in both replicates (See supplementary figure 1). This gives us confidence in the mutation calling pipeline and suggests that the mutations observed are not false positives. The high variation in mutation burden likely represents true biological variation, not technical variation.

We have added a statement to the results section stating the number of mutations per microbiopsy. While this is not equal to the number of mutations per clone, it gives the reader a better sense of how many mutations we are working with.

“We called a median of 430 single base substitutions, 39 double-base substitutions and 3 indels per microbiopsy (Methods) [..].”

A median of 357 exonic mutations (or 430 for that matter) is a very high mutation burden however, and the reviewer is correct in saying that as mutation burden increases, dN/dS ratios converge to 1 and the power to detect evidence of selection decreases. However, one can say about almost every piece of research that analyzing more samples *could have* resulted in more significant findings and those *might have* changed the interpretation of the results. It is worth noting that with nearly 1200 samples from 111 individuals, ours is one of the largest studies of somatic mutations in a non-neoplastic tissue carried out to date.

We have identified mutations under selection in psoriatic skin which occur in the same genes as are mutated in normal skin and in keratinocyte cancers. While we might be able to detect some psoriasis-specific genes by sampling more clones, the strongest selection effects would still be for the “normal” genes, like *NOTCH1*, *TP53* etc. Mutations in these genes would be more numerous and under stronger selection than any additional genes that might reach significance with greater sample size and these would dominate the selection landscape. Any potential psoriasis specific selection effects would have to be smaller and less important compared with the genes identified through studies of normal skin and skin cancers. Coming into this study, psoriasis-specific selection effects could have been stronger/more important than those in healthy skin, but our work shows that this isn't so. While we do not rule out minor effects of the disease which could become evident by increasing the sample size, we show that there are no “low hanging fruit” as in IBD or liver disease for example.

Minor issues:

5) Psoriatic plaques repeat expansion and shrinkage throughout life. Thus, non-lesional skin at the time of sampling may experience repeated inflammation and may not necessarily work as a good control. Moreover, the mutation rate therein might be overestimated because of the past exposure to inflammatory processes. In addition, lesional skin also have such history of repeated inflammation and remission.

It is true that the disease occurs in flares of inflammation and remission and that lesional areas of the skin are certain to have undergone such cycles. It is difficult for us to be certain that non-lesional skin has never been affected and indeed, this is a similar problem that we faced in our recent work in inflammatory bowel disease. To address this we've added the following paragraph to the discussion:

“Psoriasis occurs in cycles of flares and remission and although we have modeled the mutation burden as increasing as a function of disease duration, we expect that most inflammation-associated mutagenesis happens in shorter bursts during a flare up. [...]

The cycles of expansion and shrinkage of lesions that characterise psoriasis make it difficult to be certain that the non-lesional skin has never been affected in the past. If it has been, this would have the effect of lowering our estimate of the mutation burden increase associated with disease duration.”

6) While the median size of microbiopsies is <0.01mm², it would be helpful to provide an estimate of an approximate number of cells in typical microbiopsies as well.

This is a good point. Giving the number of cells would aid in the interpretation of our results. Within a microbiopsy, it is difficult to estimate the number of anucleate cells of the outermost stratum corneum. The most important parameter is probably the number of basal cells that would fit within the surface area of the microbiopsy, rather than the number of cells that would fit in a given volume.

A typical basal cell is approximately 9 micrometers in diameter. From the length of each dissection along the basal membrane and the number of z-stacks (the sections are 10 micrometers thick), we may (roughly) calculate the number of cells in the basal layer corresponding to the microdissection. We have added the following to the first paragraph of the results section:

“Assuming a diameter of 9 micrometres and vertical stratification of cells from basal to cornified layer, we estimate that a median of ~30 basal stem cells have given rise to the cells of the microbiopsies.”

8) Line 181: Patient 34 was described as an outlier. However, Fig.2A or Suppl. Fig. 3 has no Patient IDs, which makes identification of the case (Patient 34) impossible.

We have changed the legend of these figures to indicate that in both figures the leftmost panel represents the outlier patient 34.

9) Fig. 2b: The outlier case (Patient 34) should be excluded from the regression analysis.

Figure 2b only served to visualize the full raw data. No results from that regression analysis were presented in the text of the original version of the manuscript. The results from the regression analyses are all presented after excluding psoralen-related mutations and after these have been removed, the patient is no longer an outlier (see Figures 2c and 2d).

We recognize that panel b of figure 2 would be likely to cause this confusions and we have therefore removed Figure 2b and only show the linear models for mutations attributed to UV and clock-like mutational processes in the revised version of figure 2.

10) In the discussion in Lines 209-224, the result from two seminal papers (#24, #25) in IBD should be also cited and discussed.

Generally, we feel that the work presented in the two seminal IBD papers (#24 and #25) is of outstanding quality. However each study has its own strengths. Lines 209-224 relate to the increased mutation burden of SBS1/5 specifically, and for this comparison the data from Olafsson et al is superior to the other two papers. Owing to the many more mutations identified per sample through whole-genome sequencing (while refs #24 and #25 used a combination of targeted and whole-exome sequencing), we were able to identify 17 mutational signatures, including 12 SBS-signatures and estimate the mutation burden of each with much greater level of confidence than refs #24 and #25. For comparison, Kakiuchi et al identified only two signatures and neither Kakiuchi et al or Nanki et al report the burden of SBS5 in the colonic crypts.

Specifically relating to mutational signatures, Olafsson et al offers the best data of the three papers and is therefore the most relevant to the discussion in lines 209-224. We reference and discuss the results of the other two papers both in the introduction and the discussion where appropriate. There are other occasions where citing and specially highlighting the results of one of the papers is appropriate. For

example, we specifically cite Nanki et al (but not Olafsson et al) when discussing the possibility that IL-17 is cytotoxic to epithelial cells.

11) Lines 334-338: “Compared with a uniform mutational process, the psoralen-signature is 1.37 times more likely to result in splicing mutations in genes reported to be recurrently mutated in squamous cell carcinomas and normal skin. It would be especially likely to disrupt AT-AC introns, which have 5'-AT and AC-3' boundaries, in contrast to the typical 5'-GT and AG-3' splice sites.” No data are shown to support these statements.

A similar comment was raised by reviewer #2. In response, we have added the following figure as a panel to Figure 3. In the figure, we compare the probability of observing a mutation of each annotation class relative to either a uniform mutational process or relative to SBS7b (the dominant mutational process in our dataset). We plot both the values for the psoralen signature and the other major mutational signatures in the skin for comparison.

We have also changed the paragraph in question so it now reads:

“Among genes reported to be under positive selection in squamous cell carcinomas or normal skin (Methods), we counted all possible mutations that would be predicted to result in synonymous, missense, nonsense or splice site mutations. We then compared the fraction of mutations in each annotation class that would be expected for the psoralen signature and four other signatures commonly found in the skin (Figure 3f) to

the number of mutations that would be expected under a uniform background mutational process and under SBS7b, the dominant mutational process in the current study. This analysis suggests that the psoralens are less likely to result in synonymous mutations in known driver genes compared with other mutational processes affecting the skin and may be especially likely to cause splice-site mutations.”

We have added the code for carrying out this analysis to Supplementary Analysis 2 and have added the following description to the methods section:

“To assess the potential functional effects of the psoralen signature relative to other major signatures in the skin (Figure 3f), we used the `context_potential_damage_analysis()` and `signature_potential_damage_analysis()` functions of the `MutationalPatterns` package in R. Considering only the genes previously reported to be under selection in squamous cell carcinomas and/or normal skin, we counted the number of trinucleotide changes expected to give rise to the different types of mutations (Synonymous, Missense, Nonsense or Splice site). For each of the major signatures in the skin (Psoralen, SBS1, SBS5, SBS7a and SBS7b), we normalized the expected fraction of mutations falling in each annotation class by the fractions that would be expected given either a uniform mutation rate or SBS7b. We note that while this analysis gives a hint of how damaging a signature might be, it is only based on a trinucleotide mutational context and other important features such as gene expression, strandedness or the extended nucleotide context are not taken into account.

The genes we used in this analysis are *AJUBA*, *ARID2*, *ASXL1*, *CASP8*, *CDKN2A*, *FAT1*, *KMT2D*, *NOTCH1*, *NOTCH2*, *NOTCH3*, *PPM1D*, *RB1*, *RBM10*, *TP53*, *TP63*.”

12) Line 484-486: *EEF1A1* is described as being identified as a driver gene in urothelium. However, this may not support the statement “*EEF1A1* mutations are generally selected for in squamous epithelia, because urothelium is not squamous cell epithelium but is typical transitional epithelium.

We did not mean to suggest that the urothelium is a squamous epithelia but rather to suggest that since mutations in *EEF1A1* are selected for in a second normal epithelial tissue, it is likely they are also under selection in normal squamous epithelium and not a feature specific of psoriasis. To increase the clarity of this statement, we have changed it so it now reads:

“*EEF1A1* did reach significance in our previous analysis of normal urothelium from bladder cancer patients¹⁰, suggesting it too may be generally selected for in normal epithelial tissues and may not be specific to psoriasis. “

13) What is the criteria for “extensive” PUVA treatment (Line 533)? Usually, >200 sessions are commonly used to define very frequency therapy in epidemiological studies. Also, better define “limited number of PUVA” (Page 23, line 538). In Xpa deficient mice, only 1 UVB exposure is sufficient to cause skin cancer in a few months.

We have added information about the number of phototherapy sessions to the patient meta-data, assigning patients to bins with <50 sessions, 50-200 and >200 sessions. The patient line 533 refers to, patient 34, has undergone >200 sessions of PUVA therapy and this has been added to the text.

We have removed the sentence saying that a limited number of PUVA cycles are generally considered safe.

14) Many typos and errors

Page 2, line 62: “biopsie” => “biopsy”, “patients”=> “patients”

Page 4, line 1: “DNA extracted from each of the microbiopsies was whole-exome sequencing”: some words missing?

Page 16, Fig. 3: characters are barely readable in fig 3b and c.

Line 395: “failure detect” => “failure to detect”

Page 19, Fig. 4 legend: “number” rather than “umber”

Page 21 “Wehypothesize” => “We hypothesize”

We thank the reviewer for noticing and have fixed these typos.

Reviewer #2:

Remarks to the Author:

NG-A60455

Effects of psoriasis and psolaren exposure on the somatic mutation landscape of the skin

Summary

Psoriasis is a complex autoimmune disease, which often manifests as chronic inflammation of the skin. In this article, 1182 skin microbiopsies from 111 patients with psoriasis were whole exome sequenced to characterise the somatic mutations affecting psoriatic skin, in contrast with normal skin. The results indicate that there is no significant difference between the clonality composition or driver selection of normal skin and psoriatic skin. There is a difference in mutational burden of only specific mutational processes, namely those related to SBS1 and SBS5, the age-associated mutational signatures. More specifically, psoriatic skin appears to have increased mutation accumulation from these mutational processes only.

The study then moves on to characterise the mutational signature of psoralens exposure, which are chemicals used along with UV-A (PUVA) to treat psoriasis. Psoralens' mutagenic effects seem to be mostly single nucleotide variants of T>N at TpA sites. Additional characterisation of this signature helps to understand the mechanistic causes of the mutations. The authors postulate that the moderate increase in skin cancer risk for people affected by psoriasis might be due in part or fully by the PUVA treatment.

The authors provide interesting comparisons between the findings in this paper and what is already known about the somatic landscape and cellular evolution of the normal skin, as well as of a related autoimmune disease, the inflammatory bowel disease.

Overall, the article is well written, the data is of high interest and the analysis is of high standard, with appropriate evidence for the authors' claims. I have found only some minor issues that are mostly related to some of the figures, see below.

It is very encouraging to receive such positive feedback. Many thanks to the reviewer for the useful comments

Minor issues:

1. I think that more explanation is required to understand the linear model described in section 7.2 of methods. Currently, there is only a high level explanation and a supplementary R code. Neither the R code nor the current methods provide a detailed description of the mathematical model and its assumptions. Please provide these details.

We agree with the reviewer that a more comprehensive description of the model is warranted. We have re-written section 7.2 of the methods so it now reads:

We used linear mixed effect models to compare the mutation burdens of lesional and non-lesional skin. We used as response variables the estimates of the mutation burdens of SBS7 and SBS1/5 after correcting for VAF and coverage as described above. The models include fixed effects for age and the anatomical location of the biopsy from which the clone is derived. Our observations are not independent. Cell clones from the same biopsy are likely to have correlated levels of UV-exposure and some correlation is also likely to exist between biopsies taken from the same patient. To model this, we include random effects for patient and biopsy, with the effect of biopsy being nested within that of the patient.

Most embryonic mutations will be filtered as germline and so at birth the exonic mutation count of the skin is close to zero. Rather than including a random intercept in our models, we constrain the intercepts to zero. The biological interpretation being that no (exonic) somatic mutations are present in the skin at birth.

Thus, the basic model, M_0 , of mutation burden in clone i attributed to signature, S , is:

$$M_0: Burden_{S,i} = \beta_1 \times Age_i + \beta_2 \times Location_i + (\beta_3 + \beta_4) \times Patient_i Biopsy_i + \epsilon_i$$

Where β_1 is the fixed effect of age, β_2 is the fixed effect of anatomical location of the sample, $\beta_3 \sim N(0, \sigma_p)$ is the random effect of patient and $\beta_4 \sim N(0, \sigma_{pb})$ is the nested random effect for biopsy within a person. $\epsilon_i \sim N(0, \sigma_e)$ is a normally distributed error term.

We wish to know if psoriasis affects the mutation burden of keratinocytes. The ideal predictor variable would be some measure of ‘inflammation exposure’ which might comprise both the number of flare ups and the severity of each flare up. We lack this information for our patients, many of whom have had psoriasis for decades and been managed by different dermatologists during that time. Instead, we use disease duration as a proxy for disease exposure, setting the disease duration of non-lesional samples to zero before fitting the model. We note that in the event that some of the non-lesional samples are from areas of the skin which have been previously affected by the disease, our estimate of the disease effect would be smaller and this approach is therefore conservative.

To test for an effect of disease duration on the mutation burden of signature S, we define a second model, M_1 , which includes a fixed effect for disease duration, β_5 .

$$M_1: Burden_{S,i} = \beta_1 \times Age_i + \beta_2 \times Location_i + \beta_5 \times DiseaseDuration + (\beta_3 + \beta_4) \times Patient_i Biopsy_i + \epsilon_i$$

We use a likelihood ratio test to see if adding disease duration significantly improves the fit of the model. The null hypothesis is $H_0 : \beta_5 = 0$ versus $H_1 : \beta_5 \neq 0$.

We calculate a likelihood ratio statistic, D, as

$$D = -2 \times \ln\left(\frac{L(M_0)}{L(M_1)}\right) = 2 \times [LL(M_1) - LL(M_0)]$$

Where $L(M_i)$ and $LL(M_i)$ are the likelihoods and the log-likelihoods of the two models, respectively. Under the null hypothesis, D follows a chi-squared distribution with one degree of freedom, giving a P-value for the disease duration effect.

2. I might have missed it, but I didn't find a mention to the possibility that SBS1/5 might increase in psoriasis also because the psoriasis cells undergo hyperproliferation, so more cell cycles and perhaps more chances to introduce SBS1/5 mutations. Please add a few lines in the discussion to mention this possibility.

This is a good point and we agree with the reviewer that this should be discussed. Our earlier work in post-mitotic neurons and muscle indicates that cell division may not be required for accumulation of SBS1/5 related mutations. We have given the reference for this and added the following paragraph to the discussion.

“[...]The mutation burdens of substitution signatures 1 and 5 increase linearly with time and the associated mutational processes were until recently thought to be linked with cell division. Our recent work showing that these signatures accumulate linearly with time in post-mitotic cells shows that cell division is not necessary for the accumulation of SBS1/5 mutations and it remains uncertain if the observed increase in psoriasis is due to the hyperproliferation of keratinocytes in the skin or other factors relating to the inflammation.”

3. I find the histograms in Figure 1d and Supplementary Figure 1a-c confusing and I would like them to be improved. The way they are currently it is hard to tell which bar corresponds to which interval, because the x axis is continuous. This type of histogram would work for categorical data with each category labelled, but I find it unclear here. Perhaps just draw the two histograms separately.

We have replaced Figure 1d with the following figure in the hope of improving the clarity. The modified figure helps highlight that the distributions of median VAFs are highly similar between lesional and non-lesional skin.

Similarly, Supplementary figure 1 has been changed to look like this:

4. Figures 2b-d also need improvement for clarity. The main issues I find here are that the blue dots are hard to see, mostly covered by red dots. Also there is no indication of what the line and shaded region around the line are, this information should be added. Also, given the question of whether lesional and non-lesional skin are different, we should probably look at two slopes here, one for lesional and one for non-lesional. I suggest drawing lesional and non-lesional data separately and reporting some slope statistics on the figure.

We have changed figures 2b-d so they now look like this:

This splits the data by psoriasis-status, making it easier to see the blue dots. We indicate in the figure legend that the shaded regions show the 95% confidence interval of the age-effect estimate of the model.

The plots above show the same slope for lesional and non-lesional samples. Our use of linear mixed effect models means that we do not obtain two slopes, one for each type. Rather, within one model, we estimate the age effect and the disease duration effect together. The new figure 2d shows a comparison of the fixed effects for age and disease duration for three different response variables, the total mutation burden (excepting psoralen-related mutations), the UV-associated mutation burden and the mutation burden of the clock-like mutational processes responsible for SBS1/5.

5. At line 334 and following, some interesting findings about splicing mutations are reported, however there seems to be an indication of where to find these results. Is there a table with the associated values? Is there a supplementary file with the procedure? This is also missing from the methods section. Please add.

We agree with the reviewer that this point requires more attention in the manuscript. We have added the following figure as a panel to Figure 3. In the figure, we compare the probability of observing a mutation of each annotation class relative to either a uniform mutational process or relative to SBS7b (the dominant mutational process in our dataset). We plot both the values for the psoralen signature and the other major mutational signatures in the skin for comparison.

We have also changed the paragraph in question so it now reads:

“Among genes reported to be under positive selection in squamous cell carcinomas or normal skin (Methods), we counted all possible mutations that would be predicted to result in synonymous, missense, nonsense or splice site mutations. We then compared the fraction of mutations in each annotation class that would be expected for the psoralen signature and four other signatures commonly found in the skin (Figure 3f) to the number of mutations that would be expected under a uniform background mutational process and under SBS7b, the dominant mutational process in the current study. This analysis suggests that the psoralens are less likely to result in synonymous mutations in known driver genes compared with other mutational processes affecting the skin and may be especially likely to cause splice-site mutations.”

We have added the code for carrying out this analysis to Supplementary Analysis 2 and have added the following description to the methods section:

“To assess the potential functional effects of the psoralen signature relative to other major signatures in the skin (Figure 3f), we used the `context_potential_damage_analysis()` and `signature_potential_damage_analysis()` functions of the `MutationalPatterns` package in R. Considering only the genes previously reported to be under selection in squamous cell carcinomas and/or normal skin, we counted the number of trinucleotide changes expected to give rise to the different types

of mutations (Synonymous, Missense, Nonsense or Splice site). For each of the major signatures in the skin (Psoralen, SBS1, SBS5, SBS7a and SBS7b), we normalized the expected fraction of mutations falling in each annotation class by the fractions that would be expected given either a uniform mutation rate or SBS7b. We note that while this analysis gives a hint of how damaging a signature might be, it is only based on a trinucleotide mutational context and other important features such as gene expression, strandedness or the extended nucleotide context are not taken into account.

The genes we used in this analysis are *AJUBA*, *ARID2*, *ASXL1*, *CASP8*, *CDKN2A*, *FAT1*, *KMT2D*, *NOTCH1*, *NOTCH2*, *NOTCH3*, *PPM1D*, *RB1*, *RBM10*, *TP53*, *TP63*."

6. The text in Figure 3, supplementary figure 5, supplementary figure 6 some text is too small, please improve readability.

We have increased the size of the text-fonts in all these figures and made additional minor changes to the figures and their legends to improve their readability. We also discovered and fixed a bug that caused deletions of length 1 not to be correctly plotted in Supplementary Figure 5a. The error affects only the plotting code and not any other analyses of the paper and it does not change the conclusions drawn from the plot, namely that the indel mutation spectrum is the same in psoralen exposed and non-exposed individuals.

7. Figure 3f, Psoralen misspelt Poralen

Fixed.

Reviewer #3:

Remarks to the Author:

Comments on "Effects of psoriasis and psoralen exposure on the somatic mutation landscape of the skin" by Dr. Olafsson and colleagues. In this study the authors performed whole-exome sequencing of 1182 microbiopsies dissected from lesional and non-lesional epidermis from 111 patients with psoriasis to search for evidence that somatic mutations in keratinocytes may influence the disease process and to characterize the effects of the disease on the mutation landscape of the epidermis.

This study is similar to what this group has previously done on colonic mucosa in inflammatory bowel disease, published in 2020 in Cell (PMID:32697969) making the contrast with psoriasis more notable, given the very marked differences in findings between these two inflammatory diseases. There are several novel findings reported here that are of importance to the dermatology/medical field. This includes the findings that psoriasis is associated with increased mutation burden of the cell-intrinsic signatures SBS1 and SBS5 but not of UV-light. Further, the finding that despite keratinocyte hyperproliferation lesional psoriatic skin remains highly polyclonal showing no evidence of spread of clones carrying potentially pathogenic mutations, is highly interesting. Lastly, the authors detect distinct PUVA-associated mutational signature in the genomes of exposed cells that is tightly linked with transcription.

I agree with the authors these results suggest that somatic mutations in keratinocytes are unlikely to influence the pathogenesis of psoriasis.

We thank the reviewer for his/her comments and are glad to hear that the reviewer agrees with us that this work is of interest to the dermatology/medical field even though the major conclusion is that somatic mutations are unlikely to play a major role in the pathogenesis of psoriasis.

I have the following comments.

It is well established that PUVA treatment is carcinogenic in patients that have received multiple treatments. For this reason, the use of PUVA has markedly diminished and is currently rarely used, if at all, for the treatment of psoriasis. Therefore, the emphasis on this aspect of the study should be toned down, although I agree that it is very interesting. In addition, I'd expect that most of the patients that had history of PUVA treatment were older than non-PUVA treated individuals and I think the authors should address that and how the age difference in PUVA + vs PUVA- subjects, would potentially confound the data/results and their interpretation. Also, the authors calculate mutation burden of keratinocytes clones per year, but it would be of interest if the authors can show this separately for the PUVA treated group. (and not include that in the initial calculation). PUVA carcinogenesis is associated with the number of PUVA treatments, did the authors compare that in their study cohort?

This comment makes several good points which we break down below.

- 1. Use of PUVA has diminished with time**

It is important to make a distinction between PUVA treatment and the psoralen signature. While we believe that PUVA is an important source of psoralen-related mutations, we also observe the signature in several individuals with no history of PUVA treatment. These tend to be older individuals and we hypothesize that past use of psoralen-containing sunscreen (before they were banned in 1996) may be to blame.

Somatic mutations document all the mutagenic exposures of stem cells during our lifetimes and the effects of past uses of PUVA will often be visible. In our cohort, nearly 20% of patients have a history of PUVA treatment and more individuals have a history of PUVA than of UV-B for example.

2. History of PUVA may correlate with older age.

This is a great point. Indeed, the median age of patients who have a history of PUVA treatment is 54 years, compared with 46 years for patients without a history of the treatment. The same difference in age is observed between those showing the psoralen signature and those who do not. The age difference would affect only one of our analyses; the comparison of clone sizes between psoralen exposed and non-exposed individuals. The reviewer is correct that it is important to take the age of the patient into account and that our analysis had not done this. We have replaced figure 3e with the following figure, which shows that the median VAFs of microbiopsies from individuals with evidence of the psoralen signature are higher than the medians of patients without psoralen exposure, even after accounting for age ($P=0.0018$ in a linear model).

3. The mutation burden of keratinocytes should be calculated with/without PUVA

The per year mutation burden of the keratinocytes is already calculated after subtracting mutations that are attributed to the psoralen signature. These mutations do not accumulate linearly with time so it makes sense to exclude them from an analysis focused on age-related mutagenesis.

4. PUVA carcinogenesis and the number of PUVA treatments

Since the initial submission of the manuscript, we have been able to add data on roughly how many cycles of PUVA treatment the individuals in the dataset have undergone. When this is plotted against the mutation burden of the psoralen signature in individual clones, we can see a trend for increasing mutation burden with the number of PUVA cycles. However, the psoralen signature is also seen in some individuals with no history of PUVA treatment. This suggests that there are other important sources of psoralen-mutagenesis. We suggest in the manuscript that these may include past use of psoralen containing sun creams and/or consumption of fuccumarin rich foods.

When the categories above are included in a linear-mixed effects model of psoralen-signature burden (as a factor variable), this significantly improves the fit of the model ($p=1.2E-6$, likelihood ratio test) although only the group with >200 cycles of PUVA have a significantly higher burden of the psoralen signature than the group without PUVA history (616 extra mutations per clone. 407-826 95% CI).

We have added the figure above as a panel to figure 3 and added the following sentence to the results section of the manuscript:

“We found that including the number of PUVA cycles as a covariate in a model of the psoralen-signature mutation burden significantly improved the fit of the model ($P=1.2 \times 10^{-6}$, likelihood ratio test) and although only the group with >200 cycles of PUVA had a higher psoralen mutation burden than those without history of PUVA treatment (616 extra (exonic) mutations per clone. 407-826 95% CI), we observed a trend for increased mutation burden with the number of PUVA cycles (Figure 3b)”

I’m surprised that the authors do not address potential mutations burden and analyses for patients treated with UVB, which is the most commonly used phototherapy used to treat psoriasis patients, and it’s surprising that this is not addressed here at all. Also, from reading the manuscript it is not clear how, and if, the authors distinguish natural UV exposure from UV-treatment related exposure. In addition, and related to that, do the authors see site-specific differences in their UV signatures? i.e., chronically sun exposed sites (i.e., legs/arms) vs. more sun-protected sites (flank/abdomen sites)?

We agree with the reviewer that these are interesting questions that should be better addressed in the manuscript. Since the initial submission, we have been able to add clinical data on the number of cycles of UVB and other therapies. We show these data in revised supplementary figure 1, panels b and c.

In our dataset, history of PUVA treatment is slightly more common than history of UVB. For both treatment types, most people with a history of the treatment have received fewer than 50 cycles of phototherapy. We show the psoralen mutation burden as a function of the number of PUVA cycles in the previous response. We have created a similar plot for UV-B. Only a single patient has had 51-200 cycles of UVB and none have had >200 cycles. This individual did show a higher-than-average burden of SBS7b, but the difference does not reach statistical significance.

The figure and the code to reproduce it has been made available in a new supplementary analysis document, which focuses specifically on the effects of treatments (see below).

“Unsurprisingly, the most abundant signature is SBS7b, which has been attributed to UV-exposure. This signature accounts for 80% of the mutations in the dataset. We also found the UV-related signature SBS7c, but it only accounts for 0.14% of the mutations. In agreement with previous studies^{16,22}, we observed large variation in UV-associated mutation burden between cells 1-2mm apart in the tissue. 26 of the patients in our cohort have a history of phototreatment with UV-B (Supplementary figure 1b and 1c). We did not find those individuals to have a significantly higher burden of SBS7b and SBS7c than patients without history of UV-B treatment (Supplementary Analysis 3). However, we note that most patients had undergone relatively few (<50) cycles of UV-B treatment and the large variation in UV-related mutation burden reduces the statistical power to detect an effect of this treatment. Only one individual had received >200 cycles of UV-treatment. Although clones from this individual did show a higher-than-average burden of UV-related mutations (Supplementary Analysis 3), this was not sufficient to show a statistical relationship between history UV-B treatment and UV-related mutation burden. “

What about other treatments? Many of these patients may have been treated with methotrexate and other antimetabolite treatments that will directly interfere with DNA synthesis. This should be addressed by the authors as well.

Mutagenic drugs, such as many chemotherapy agents, typically leave a characteristic mutational signature on the genomes of exposed cells. The fact that we saw no (credible) novel mutational signatures other than the psoralen signature suggested to us that other treatments were unlikely to affect the mutagenic landscape of the skin, which is why we didn't pursue this line of thought initially.

We have added a supplementary analysis focusing on the effects of different treatments. Therein we show that methotrexate and topical steroids have no effect on the mutational spectra, the mutation burden or on the clonal composition of the tissue. The document also includes the analyses of PUVA and UV-B described in our earlier responses.

The mutational signature of azathioprine use has been described in squamous cell carcinomas (<https://www.nature.com/articles/s41467-018-06027-1>) but none of the patients in our cohort have a history of azathioprine use.

In addition to the paragraph on UV-B in our reply to the previous comment, we have added the following statement to the section on mutagenic processes in the psoriatic skin:

We found no evidence that other treatments, including methotrexate or topical steroid use, cause somatic mutations in the skin (Supplementary Analysis 3).

Line 506. I'm not sure that the argument that both IBD and psoriasis are Th17 mediated chronic inflammatory diseases as the role of IL-17 in IBD seems to be very different, with anti-IL-17 drugs sometimes unmasking or triggering IBD, while highly effective in treating psoriasis, suggesting a completely different role of IL-17A in those two diseases.

Indeed the role of IL-17 seems to be very different between IBD and psoriasis. One of the things we were highly interested in when conceiving of the project was if clones carrying somatic mutations in the IL-17 pathway would enjoy a similar selection advantage in psoriasis-affected skin to that seen in IBD-affected colonic mucosa. This is not the case and the point of the comparison in line 506 was to highlight this difference. We have re-phrased this paragraph for clarity so it now reads:

“In IBD, mutations in immune-related genes are under positive selection and these may play a role in the pathogenesis of the disease and/or enable cells to escape the cytotoxic effects of IL-17. Mutations in genes of the IL-17 pathway are not positively selected in psoriasis, underscoring the seemingly different role of this cytokine in these two diseases. It is worth noting that while IL-17 inhibitors work well for psoriasis, clinical trials of these drugs in IBD have either shown the drugs to not be effective or that they worsen the disease. We did not find evidence of positive selection of mutations in any immune related or psoriasis-related genes and we conclude that although somatic mutations may offer insights into the cellular dynamics that accompany disease, somatic mutations in the epidermis are unlikely to play a major role in the pathogenesis of psoriasis. Importantly, our data does not exclude a role for somatic mutations in lymphocytes in psoriasis pathogenesis(Goodnow 2007). Emerging

data from other autoimmune diseases suggests that somatic mutations may sometimes enable lymphocytes to bypass tolerance checkpoints, resulting in expansion of autoreactive clones (Singh et al. 2020; Masle-Farquhar et al. 2022). Determining whether similar mechanisms are at play in psoriasis is an important direction of future work. “

Line 541. I'm not aware of psoralens being still used in sunscreens as this practice has been banned in most countries more than 20 years ago. Therefore, this argument has low relevance and should be removed. I agree with the authors about the potential in regards to consumption of furocoumarin-rich foods and this should be kept although this in itself is not a focus of the current manuscript and I do not think the authors have sufficient data on this particular aspect of fourcoumarins to make any definite statements on that .

It is true that psoralens are not used in sunscreens today. In line 541 we meant that **past** use of psoralen-containing sunscreen is a possible reason for why one might observe the signature today. The mutational profile of a cell contains information about all mutagens the cell has been exposed to during the lifetime of the individual. Great many people who used psoralen-containing sunscreens are still alive and will develop skin cancers in the coming years. Indeed, assuming that people are more likely to use sunscreens (particularly those advertised as “tanning-enhancers”) early in life, the people who used psoralen containing sunscreens the most may only now be reaching the age at which most skin cancers manifest. We therefore respectfully disagree with the reviewer that this argument has low relevance but have added the word “past” to the sentence to clarify in the discussion that we do not mean that people are still actively using psoralen-containing sunscreens. It now reads:

“While psoralen exposure in the context of psoriasis is particularly likely to occur during PUVA-therapy, other means of exposure include the **past** use of sunscreen or tanning lotions containing psoralens or even the consumption of furocoumarin-rich foods.”

We have also improved on our explanation of what mutational signatures are by changing the second paragraph of the Mutagenic processes results section so it now reads:

“The somatic mutations found in a clone of skin cells reflect the effects of all the mutagenic processes the cell has been exposed to over the lifetime of the individual. Different mutational processes affect DNA in different way such that each leaves a characteristic pattern, a mutational signature, within the genome, commonly represented by a 96-class profile where the mutated pyrimidine base is plotted together

with the bases immediately 3' and 5' of the mutated base (Alexandrov et al. 2013; Alexandrov et al. 2020).

To determine which mutational processes have been active in the skin over the lifetime of our psoriasis patients, we used a Bayesian hierarchical Dirichlet process [...]"

Regarding the furocoumarin-rich foods, although there have been several epidemiological reports linking consumption of furocoumarin-rich foods with skin cancer rate, many remain unconvinced of the causal relationship. During the initial submission, we were not in a position to make any statements on this work. We have since looked for evidence of the psoralen-signature in two skin cancer cohorts and only found the signature to be present in a single case. An individual with psoriasis and a history of PUVA treatment. We have changed the paragraph in the discussion on furocoumarins so it now reads:

While psoralen exposure in the context of psoriasis is particularly likely to occur during PUVA-therapy, other means of exposure include the past use of sunscreen or tanning lotions containing psoralens or even the consumption of furocoumarin-rich foods. It has been hypothesised that orally ingested furocoumarins increase risk of skin cancers⁵⁶⁻⁵⁹. We looked for evidence of the psoralen signature both in TCGA melanoma samples and in skin cancers from the HMF and found evidence of the signature only in a psoriasis patient with a history of PUVA treatment. This does not rule out a mutagenic effect of furocoumarins from diet, but suggests that they are rarely a major source of mutations in the populations from which these cohorts are drawn. We propose that testing for a relationship between furocoumarin consumption and the psoralen mutational signature in sun exposed skin offers a mechanistic way to test the hypothesis that furocoumarin rich foods cause skin cancers.

Patient demographics are incomplete and need to be expanded – also the format as listed is not easily approachable and should be revised in a table format. Treatment only includes PUVA but should also include other treatments, in particular UVB and MTX as those are both potential mutation inducing agents.

We feel that for future users of the dataset, it is important to make the clinical information available on a per-patient level. However, we agree with the reviewer that this makes it more difficult for the reader

to get a quick sense of the patient demographics. We have kept Supplementary Table 1, which lists the demographics on a per-patient basis, but have added to this information about treatment histories with PUVA, UVB, methotrexate and topical steroids.

As ours is not a case-control study, we feel a table would not be too useful for showing the patient demographics. It would essentially just be a list of numbers. Instead, to provide the reader with a better overview of the patient demographics, we have added three panels to Supplementary Figure 1 which show the disease duration as a function of patient age, broken down by sex and histograms for both of these variables. Supplementary figure 1 also now includes data on treatment history for four of the most common treatments, methotrexate, PUVA, topical steroids and UV-B. Other treatments were used too sporadically to be powered in a statistical analysis.

Minor comment**Line 464....lacks space between two words "Wehypothesize"**

Fixed

Decision Letter, first revision:

14th Mar 2023

Dear Dr Anderson,

I hope you're well.

Your Article, "Effects of psoriasis and psoralen exposure on the somatic mutation landscape of the skin." has now been seen by your three original 3 referees. We are interested in the possibility of publishing your study in Nature Genetics, but would like to consider your response to these concerns in the form of a revised manuscript before we make a final decision on publication.

You'll see that Reviewer #1 continues to have concerns about the possibility of multiple clones within a single LCM sample. In their opinion, this undermines confidence in your estimation of the mutation rate, and also for the construction of phylogenetic trees. We therefore invite you to revise your manuscript taking into account all their comments. Please highlight all changes in the manuscript text file. At this stage we will need you to upload a copy of the manuscript in MS Word .docx or similar editable format.

*2) If you have not done so already please begin to revise your manuscript so that it conforms to our Article format instructions, available

[here](http://www.nature.com/ng/authors/article_types/index.html).

*3) Include a revised version of any required Reporting Summary:

[redacted]

We hope to receive your revised manuscript within four to eight weeks. If you cannot send it within this time, please let us know.

Sincerely,

Safia Danovi
Editor
Nature Genetics

Reviewers' Comments:

Reviewer #1:
Remarks to the Author:

In their revised manuscript, the authors address most of the concerns except for the first one. The critical issue is that low VAF values in many samples suggest that this study does not capture mutations in single clones. The authors' reply is not acceptable on this point.

Major comments:

1) Estimation of mutation rate

General comment:

Given a median VAF of 0.25, the clonal structure of many clones cannot uniquely be determined, even using phylogenetic analysis. This reviewer checked through the mutation clusters and the estimated cell fractions summarized in cluster_cell_fractions at <https://data.mendeley.com/datasets/rfcy88sb9s>, where many mutation clusters showed cell fractions < 0.5. For those cluster, it is not always possible to determine whether they represented a single clone or a collection of clones having similar cell fractions. The pigeonhole principle may not always help to discriminate.

The authors constructed phylogenetic trees as if it had only 1 clone, which should lead to overestimation of terminal branches. For example, it is not sure whether the red cluster in Fig. 1F represents a single clone or two clones with similar VAFs.

This reviewer agrees that the mutational burden in microbiopsies increase with aging. However, it is not acceptable to calculate the mutation rate as the slope in Fig. 2b, because the number of mutations may not represent that in a single clone. In fact, the mutation rate of 14.6 /exome/year seems to be too high. Moreover, there is no rational reason that similar VAF distributions between lesional and non-lesional samples suggest similar clonal structure therein. Thus, the authors claims regarding mutation rates in single clone are not based on experimental evidence and are not acceptable.

Specific comments:

- However, the median VAFs of lesional and non-lesional skin are 0.25 and 0.26, respectively. This is similar to the median VAFs reported in many cancer sequencing studies and we thus do not believe the distribution of VAFs is "skewed to very low values".

> Low median VAFs in cancer studies are nothing to do with those in the current study. The former is related to contamination of non-cancer tissues and/or the presence of multiple subclones. By contrast, the low VAFs in their study are most likely due to the presence of multiple independent clones, because the authors carefully microdissected to exclude non-epithelial cells.

- A tail of low-VAF mutations is missed in all studies, cancer or normal tissue, that do not use some variant of single-cell or single-molecule sequencing. Although the mutation rate we have estimated is subject to some uncertainty owing to low VAFs in a small fraction of the microbiopsies (just as in cancer sequencing studies), it is nevertheless the best estimate in the literature.

> According to Fig. 1c, a substantial fraction of microbiopsies showed low VAFs <0.25. These samples are thought to comprise multiple clones. The mutation rate determined based on these multiclonal samples is therefore not reliable and could be misleading.

-Past studies of normal skin have used targeted sequencing panels and sequenced biopsies of skin two orders of magnitude larger than those in the current study. The samples have thus been a mixture of very many clones and, unfortunately, these studies have not been able to assess the mutation burden of the skin at all.

> The same criticism is applied to the current study, which is also suffering from a mixture of multiple clones.

While it is true that the low VAFs in a minority of biopsies add some uncertainty to the phylogenetic

tree building for some individuals, we have two advantages over traditional cancer sequencing studies which help overcome the problem.

> Figure 1C shows that more than half samples had lower VAFs < 0.25.

Secondly, the very small size of the microbiopsies (typically <100um by <60um) means that there is very little room for multiple clones to co-exist and yet for each to be prevalent enough to pass the detection limit of this study.

> This may not be the case. Depending on clone size, several clone can coexist, for example, 2 or 3 independent clones having cell fractions of 0.5, 0.3, and 0.2 can occur.

Reviewer #2:

Remarks to the Author:

The authors have carefully addressed all comments from all authors, improving the manuscript substantially. The additional supporting material, including more explanation of the mathematical models used and detailed supplementary data provided via the external mendeley link (e.g., phylogeny reconstruction for each patient and VAF distribution of each microsection) was particularly useful to address some of the reviewer's comments. The authors have also clarified some of the limitations of their approach.

I consider my comments to be fully addressed and believe this article to be suitable for publication in Nature Genetics. I am not aware of other published work that could compromise the novelty of this article.

Reviewer #3:

Remarks to the Author:

The authors have done a good job of addressing my critiques/concerns as well as that of others. I have no additional comments.

Author Rebuttal, first revision:

Reviewer #1:

Remarks to the Author:

In their revised manuscript, the authors address most of the concerns except for the first one. The critical issue is that low VAF values in many samples suggest that this study does not capture mutations in single clones. The authors' reply is not acceptable on this point.

Major comments:

1) Estimation of mutation rate

General comment:

Given a median VAF of 0.25, the clonal structure of many clones cannot uniquely be determined, even using phylogenetic analysis. This reviewer checked through the mutation clusters and the estimated cell fractions summarized in cluster_cell_fractions at <https://data.mendeley.com/datasets/rfcy88sb9s>. [data.mendeley.com] where many mutation clusters showed cell fractions < 0.5. For those cluster, it is not always possible to determine whether they represented a single clone or a collection of clones having similar cell fractions. The pigeonhole principle may not always help to discriminate.

The authors constructed phylogenetic trees as if it had only 1 clone, which should lead to overestimation of terminal branches. For example, it is not sure whether the red cluster in Fig. 1F represents a single clone or two clones with similar VAFs.

This reviewer agrees that the mutational burden in microbiopsies increase with aging. However, it is not acceptable to calculate the mutation rate as the slope in Fig. 2b, because the number of mutations may not represent that in a single clone. In fact, the mutation rate of 14.6 /exome/year seems to be too high. Moreover, there is no rational reason that similar VAF distributions between lesional and non-lesional samples suggest similar clonal structure therein. Thus, the authors claims regarding mutation rates in single clone are not based on experimental evidence and are not acceptable.

Specific comments:

- However, the median VAFs of lesional and non-lesional skin are 0.25 and 0.26, respectively. This is similar to the median VAFs reported in many cancer sequencing studies and we thus do not believe the distribution of VAFs is “skewed to very low values”.

> Low median VAFs in cancer studies are nothing to do with those in the current study. The former is related to contamination of non-cancer tissues and/or the presence of multiple subclones. By contrast, the low VAFs in their study are most likely due to the presence of multiple independent clones, because the authors carefully microdissected to exclude non-epithelial cells.

- A tail of low-VAF mutations is missed in all studies, cancer or normal tissue, that do not use some variant of single-cell or single-molecule sequencing. Although the mutation rate we have estimated is subject to some uncertainty owing to low VAFs in a small fraction of the microbiopsies (just as in cancer sequencing studies), it is nevertheless the best estimate in the literature.

> According to Fig. 1c, a substantial fraction of microbiopsies showed low VAFs <0.25 . These samples are thought to comprise multiple clones. The mutation rate determined based on these multiclonal samples is therefore not reliable and could be misleading.

-Past studies of normal skin have used targeted sequencing panels and sequenced biopsies of skin two orders of magnitude larger than those in the current study. The samples have thus been a mixture of very many clones and, unfortunately, these studies have not been able to assess the mutation burden of the skin at all.

> The same criticism is applied to the current study, which is also suffering from a mixture of multiple clones.

While it is true that the low VAFs in a minority of biopsies add some uncertainty to the phylogenetic tree building for some individuals, we have two advantages over traditional cancer sequencing studies which help overcome the problem.

> Figure 1C shows that more than half samples had lower VAFs < 0.25 .

Secondly, the very small size of the microbiopsies (typically $<100\mu\text{m}$ by $<60\mu\text{m}$) means that there is very little room for multiple clones to co-exist and yet for each to be prevalent enough to pass the detection limit of this study.

> This may not be the case. Depending on clone size, several clones can coexist, for example, 2 or 3 independent clones having cell fractions of 0.5, 0.3, and 0.2 can occur.

These comments all boil down to a single issue, that the estimation of the mutation burden may not be accurate given the inclusion of clones with low VAFs. We have reconstructed all phylogenetic trees after pruning away branches where there is doubt about the validity of the pigeonhole principle. We now retain nested clusters only if the sum of the cellular fraction estimates exceeds 1. Un-nested clusters (i.e. branches of the phylogenetic tree consisting of a single cluster) are retained if the median VAF of the cluster is larger than 0.3.

This pruning of terminal branches of the phylogenetic trees naturally reduces the mutation rate estimates from our linear mixed effect. In the table below, we provide a summary of model coefficients before and after the tree-pruning:

	Before pruning (N=1100 tree tips)		After pruning (N=836 tree tips)	
	Beta	P	Beta	P
Total SBS per year (excluding psoralen mutations)	14.6 (10.1-19.0)	4.2E-9	9.6 (6.5-12.7)	2.0E-8
UV SBS per year	13.6 (9.1-18.1)	3.2E-8	9.0 (5.8-12.1)	1.5E-7
SBS 1/5 per year of age	0.69 (0.53-0.85)	9.1E-14	0.52 (0.39-0.66)	2.3E-11
SBS 1/5 per year of disease	0.16 (0.04-0.29)	0.012	0.07 (-0.03-0.18)	0.17

The most important change is that the effect of disease duration on the burden of SBS1/5 (bold in the table) is no longer significant.

We have made the following changes to the manuscript:

1. Removed from the abstract the following statement about increased mutation burden of SBS1/5 in psoriasis: “We show that psoriasis is associated with increased mutation burden of the cell-intrinsic signatures SBS1 and SBS5 but not of UV-light, which remains the dominant mutagen in psoriatic skin.”

2. Added the following paragraph to the results section on the linear mixed-effects modeling:

“Some of the mutation clusters consisted of groups of mutations with VAFs too low for the pigeonhole principle to be incontrovertible. The calculations above assume that in such cases, the mutations all derive from a single sub-clone. However, there is a risk that the mutation burden represents not the burden of a single clone but the sum of the mutation burden for a collection of clones with similar cell fractions across all microbiopsies. This would lead to an over-estimation of the mutation rate for terminal branches of the phylogenetic trees. We performed pruning of the phylogenetic trees, retaining only branches representing nested clusters if the sum of the VAFs was greater than 1. For branches that represent single clusters (with no nesting), we pruned branches with $VAF < 0.3$. Repeating our linear mixed effects modelling, we estimate from the pruned trees that the total mutation rate excluding psoralens is 9.6 mutations per exome per year (9.6 (6.5-12.7, 95% CI), $P=2.0E-8$), the UV-specific mutation rate is 9 mutations per exome per year (9.0 (5.8-12.1, 95% CI), $P=1.5E-7$), the age effect of SBS 1/5 is estimated to be 0.69 mutations per exome per year (0.52 (0.39-0.66, 95% CI), $P=2.3E-11$) while the disease duration effect is estimated to be 0.07 mutations per exome per year (0.07 (-0.03-0.18, 95% CI), $P=0.17$). As expected, all effects become smaller when terminal branches of the trees are pruned, but the most important difference is that the effects of disease duration on the burden of SBS1/5 is not significantly different from zero in this analysis.”

3. We have removed two paragraphs discussing the increased burden of SBS1/5 from the discussion section and replaced them with the following:

“Although the effects of UV-light (SBS7) unsurprisingly dominate in the skin, other mutational processes also affect keratinocytes and may be affected by disease. There is evidence from other diseases that

inflammation is associated with increased mutation burden of the clock-like signatures SBS1 and SBS5, which represent cell-intrinsic processes and are found in all cells of the body. For example, we previously estimated an effect of 36 mutations per genome for each year of disease duration in IBD²³. Here we found that disease duration of psoriasis, our best proxy for inflammation exposure, is associated with increased mutation burden of SBS1/5. We estimate that psoriasis disease duration is associated with an additional 0.16 mutations per exome per year, which is approximately a quarter of the age effect of these signatures. However, this estimate is dependent on certain assumptions about the clonal structures of the microbiopsies and more data will be needed to corroborate this potential effect. If the burden of SBS1/5 is increased in psoriasis, it will only result in a modest increase in the total mutation burden of keratinocytes, as SBS1/5 explains a small fraction of the total mutation burden of keratinocytes compared with UV-related signatures.”

Reviewer #2:**Remarks to the Author:**

The authors have carefully addressed all comments from all authors, improving the manuscript substantially. The additional supporting material, including more explanation of the mathematical models used and detailed supplementary data provided via the external mendeley link (e.g., phylogeny reconstruction for each patient and VAF distribution of each microsection) was particularly useful to address some of the reviewer’s comments. The authors have also clarified some of the limitations of their approach.

I consider my comments to be fully addressed and believe this article to be suitable for publication in Nature Genetics. I am not aware of other published work that could compromise the novelty of this article.

We thank the reviewer very much for their kind words and for all the work they have done in the publication process.

Reviewer #3:

Remarks to the Author:

The authors have done a good job of addressing my critiques/concerns as well as that of others. I have no additional comments.

We thank the reviewer for their work in the publication process, it is very much appreciated.

Decision Letter, second revision:

19th Jun 2023

Dear Dr. Anderson,

Thank you for submitting your revised manuscript "Effects of psoriasis and psoralen exposure on the somatic mutation landscape of the skin." (NG-A60455R2). It has now been seen by the original referees and their comments are below. The reviewers find that the paper has improved in revision, and therefore we'll be happy in principle to publish it in Nature Genetics, pending minor revisions to satisfy the final requests of Reviewer #1 and to comply with our editorial and formatting guidelines.

Sincerely,

Safia Danovi
Editor
Nature Genetics

Reviewer #1 (Remarks to the Author):

In their revised manuscript, Olafsson et al. recalculate the mutation rates taking into account only those clones showing >0.3 VAF. It provided more conservative estimations for psoriasis samples, which were not significantly different from those in normal skin. Meanwhile, pruning of terminal branches of the phylogenetic trees surely underestimate the mutation rate, because the truncal clones predated the time of sample collection. So it is still not sure if psoriasis can affect the mutation rate... However, this reviewer agrees that this is an interesting study, while it would be nice for the authors to provide some discussion about the possibility of underestimation of the mutation rate. Otherwise, no further concerns about the manuscript. Thank you for this opportunity to review this excellent study.

Author Rebuttal, second revision

Reviewer #1 (Remarks to the Author):

In their revised manuscript, Olafsson et al. recalculate the mutation rates taking into account only those clones showing >0.3 VAF. It provided more conservative estimations for psoriasis samples, which were not significantly different from those in normal skin. Meanwhile, pruning of terminal branches of the phylogenetic trees surely underestimate the mutation rate, because the truncal clones predated the time of sample collection. So it is still not sure if psoriasis can affect the mutation rate... However, this reviewer agrees that this is an interesting study, while it would be nice for the authors to provide some discussion about the possibility of underestimation of the mutation rate. Otherwise, no further concerns about the manuscript. Thank you for this opportunity to review this excellent study.

We thank the reviewer for his/her useful comments which improved the study. We have added the following sentence to the manuscript:

"We note that pruning of the phylogenetic trees can only ever shorten the tips of the phylogenetic trees and likely results in an underestimation of the mutation rate. An accumulation of 9.6 mutations per exome per year in the skin should therefore be considered a conservative lower bound estimate."

Final Decision Letter:

20th Sep 2023

Dear Dr Anderson,

I am delighted to say that your manuscript "Effects of psoriasis and psoralen exposure on the somatic mutation landscape of the skin." has been accepted for publication in an upcoming issue of Nature Genetics.

Your paper will be published online after we receive your corrections and will appear in print in the next available issue. You can find out your date of online publication by contacting the Nature Press Office (press@nature.com) after sending your e-proof corrections. Now is the time to inform your Public Relations or Press Office about your paper, as they might be interested in promoting its publication. This will allow them time to prepare an accurate and satisfactory press release. Include your manuscript tracking number (NG-A60455R3) and the name of the journal, which they will need when they contact our Press Office.

Please note that *Nature Genetics* is a Transformative Journal (TJ). Authors may publish their research with us through the traditional subscription access route or make their paper immediately open access through payment of an article-processing charge (APC). Authors will not be required to make a final decision about access to their article until it has been accepted. [Find out more about Transformative Journals](https://www.springernature.com/gp/open-research/transformative-journals)

Authors may need to take specific actions to achieve [compliance](https://www.springernature.com/gp/open-research/funding/policy-compliance-faqs) with funder and institutional open access mandates. If your research is supported by a funder that requires immediate open access (e.g. according to [Plan S principles](https://www.springernature.com/gp/open-research/plan-s-compliance)) then you should select the gold OA route, and we will direct you to the compliant route where possible. For authors selecting the subscription publication route, the journal's standard licensing terms will need to be accepted, including [a href="https://www.nature.com/nature-portfolio/editorial-policies/self-archiving-and-license-to-publish"](https://www.nature.com/nature-portfolio/editorial-policies/self-archiving-and-license-to-publish). Those licensing terms will supersede any other terms that the author or any third party may assert apply to any version of the manuscript.

An online order form for reprints of your paper is available at [a href="https://www.nature.com/reprints/author-reprints.html"](https://www.nature.com/reprints/author-reprints.html). Please let your coauthors and your institutions' public affairs office know that they are also welcome to order reprints by this method.

If you have not already done so, we invite you to upload the step-by-step protocols used in this manuscript to the Protocols Exchange, part of our on-line web resource, natureprotocols.com. If you complete the upload by the time you receive your manuscript proofs, we can insert links in your article that lead directly to the protocol details. Your protocol will be made freely available upon publication of your paper. By participating in natureprotocols.com, you are enabling researchers to more readily reproduce or adapt the methodology you use. [Natureprotocols.com](http://natureprotocols.com) is fully searchable, providing your protocols and paper with increased utility and visibility. Please submit your protocol to <https://protocolexchange.researchsquare.com/>. After entering your [nature.com](http://www.nature.com) username and password you will need to enter your manuscript number (NG-A60455R3). Further information can be found at <https://www.nature.com/nature-portfolio/editorial-policies/reporting-standards#protocols>

Sincerely,

Safia Danovi
Editor
Nature Genetics